# Chitosan-Based Scaffolds for the Treatment of Myocardial Infarction: A Systematic Review

**DOI:** 10.3390/molecules28041920

**Published:** 2023-02-17

**Authors:** Bryan Beleño Acosta, Rigoberto C. Advincula, Carlos David Grande-Tovar

**Affiliations:** 1Grupo de Investigación de Fotoquímica y Fotobiología, Química, Universidad del Atlántico, Carrera 30 Número 8-49, Puerto Colombia 081008, Colombia; 2Department of Chemical and Biomolecular Engineering, University of Tennessee, Knoxville, TN 37996, USA; 3Center for Nanophase Materials Sciences (CNMS), Oak Ridge National Laboratory, Oak Ridge, TN 37830, USA

**Keywords:** biopolymers, chitosan scaffolds, cardiac tissue engineering, natural polysaccharide, heart attack

## Abstract

Cardiovascular diseases (CVD), such as myocardial infarction (MI), constitute one of the world’s leading causes of annual deaths. This cardiomyopathy generates a tissue scar with poor anatomical properties and cell necrosis that can lead to heart failure. Necrotic tissue repair is required through pharmaceutical or surgical treatments to avoid such loss, which has associated adverse collateral effects. However, to recover the infarcted myocardial tissue, biopolymer-based scaffolds are used as safer alternative treatments with fewer side effects due to their biocompatibility, chemical adaptability and biodegradability. For this reason, a systematic review of the literature from the last five years on the production and application of chitosan scaffolds for the reconstructive engineering of myocardial tissue was carried out. Seventy-five records were included for review using the “preferred reporting items for systematic reviews and meta-analyses” data collection strategy. It was observed that the chitosan scaffolds have a remarkable capacity for restoring the essential functions of the heart through the mimicry of its physiological environment and with a controlled porosity that allows for the exchange of nutrients, the improvement of the electrical conductivity and the stimulation of cell differentiation of the stem cells. In addition, the chitosan scaffolds can significantly improve angiogenesis in the infarcted tissue by stimulating the production of the glycoprotein receptors of the vascular endothelial growth factor (VEGF) family. Therefore, the possible mechanisms of action of the chitosan scaffolds on cardiomyocytes and stem cells were analyzed. For all the advantages observed, it is considered that the treatment of MI with the chitosan scaffolds is promising, showing multiple advantages within the regenerative therapies of CVD.

## 1. Introduction

Cardiovascular disease (CVD) is commonly associated with coronary artery disease (CAD) and arteriosclerotic cardiovascular disease (ACD) [1]. Myocardial infarction (MI) is a common complication of these heart diseases. Myocardial infarction occurs with the sudden blockage of one or several coronary arteries, cutting off the blood flow in an area of the heart and causing acute pain, cell necrosis and even heart failure leading to sudden death [2]. It is well known that the heart has a low cellular regenerative capacity in response to an ischemic event or injury [3]. This natural condition denotes myocardial infarction as one of the leading causes of death, affecting more than seven million people worldwide, with 35% experiencing a fatal heart attack followed by sudden death [4].

Traditionally, myocardial infarction is treated with antiplatelet drugs, anticoagulants, β-blockers and invasive surgical reperfusion therapies, such as percutaneous coronary intervention (PCI) and coronary artery bypass grafting (CABG). However, these treatments have many limitations such as long waiting times for the patient, which, alongside other pre-existing medical conditions, increase the mortality rate [5,6].

Tissue engineering studies the application of artificial or biological matrices for transplanting cells cultured in vivo or in vitro (traditional tissue engineering), using cell expansion on the tissue (cell therapy) or using the closed-loop method. Each cell implantation strategy allows for restoring the tissues and functions of failing organs [7]. Regenerative tissue engineering using scaffolds based on natural or synthetic biopolymers [8,9] has been successfully applied in the regeneration of osseous [10], epithelial [11] and muscular [12] tissues.

Some biopolymers have been applied to study their contribution to tissue reconstruction. For example, silk protein-based scaffolds supported and facilitated human soft tissue reconstruction [13]. Therefore, the Food and Drug Administration Agency (FDA) approved the marketing of these biomaterials due to the excellent results obtained in clinical studies [13]. Similarly, scaffolds based on polylactic acid (PLA) were approved by the FDA due to their ability to stimulate cartilage repair and replacement [14].

Chitosan is a natural polymer extracted from chitin (the second most abundant polysaccharide in nature) [15]. Chitosan can be chemically modified to obtain customized properties. Its low cytotoxicity, biocompatibility, biodegradability and antimicrobial capacity have promoted its use in scaffold manufacturing [16,17]. Currently, the role of chitosan in regenerative medicine is also being studied [18]. The study by Castro et al. demonstrated that chitosan scaffolds incorporated with polyvinyl alcohol and carbon nano onions were resorbed in the subdermal tissue of Wistar rats without an aggressive inflammatory response during the implantation period [19]. In addition, chitosan with collagen [20], gelatin [21], dextran [22], alginate [23] and other biodegradable polymers have been successfully applied in regenerating dermal tissue [23,24,25].

Since the early 2000s, chitosan has been studied in pharmacological therapies to prevent CVD [26]. In 2013, the first study on scaffolds with a nanofibrous structure of chitosan/fibronectin in the co-cultures for cell transplantation in an infarcted tissue was reported [27]. This publication prompted polysaccharides to be applied in damaged heart tissue as a strategy for myocardial treatment without eliciting an aggressive immune response. Nevertheless, pure chitosan scaffolds did not generate promising results due to their crystallinity, rigidity and poor solubilization at the physiological pH. A better solution was integrating chitosan with other components to overcome the limitations [28]. Some specific properties that were achieved when chitosan was incorporated with conductive materials include the scaffold’s biocompatibility, accelerated cardiomyogenesis of the stem cells and the achievement of the electrical synchronization between the heart and the cells implanted in the scaffold, as well as a better protein uptake [29]. The lack of an aggressive immune response is possible due to the ability of modifying the chemical composition to meet the requirements of the tissue [30]. The main requirement for the application of the scaffolds in cardiac tissue is to have mechanical properties similar to those of the myocardium without causing high tension in the walls of the heart or aggressive immune responses [8]. In order to emulate the properties of the native myocardial tissue, chitosan has been modified with other polysaccharides, proteins or biodegradable polymers to reduce the stress on the cardiac walls, improve the mechanical properties, increase the biocompatibility and improve the biomechanical functions of the infarcted heart [31,32]. For example, the combination of chitosan with polypyrrole (PPi) and polycaprolactone (PCL) has been reported for the design of the scaffolds for cardiac cell regeneration [29].

Multiple advantages have been observed when applying chitosan scaffolds with other biopolymers in cardiac tissue regeneration. For example, it was observed that the chitosan and alginate scaffolds promoted rapid bioadsorption in the infarcted tissue, improving the regenerative properties of the heart muscles [33]. Currently, the exact mechanism of how the scaffolds manage the regeneration of the infarcted tissue remains unrevealed. However, Nomura et al. [34] reconstructed the cardiomyocyte remodeling and elucidated the cardiomyocyte gene programs encoding, morphological and functional characteristics in cardiac hypertrophy and failure, correlating single-cardiomyocyte transcriptome with a cell morphology, epigenomic state and heart function. In addition, it is essential to understand and elucidate the gene programs in stimulating cardiomyocytes for cardiac cell regeneration using the scaffolds on the infarcted myocardium [34,35].

Due to the potential of chitosan-based scaffolds in the treatment of the infarcted myocardial tissue, this systematic literature review of the last five years analyzed the ability for the chitosan-based scaffolds to stimulate the reconstruction of the infarcted myocardium, discussing the results obtained and their perspectives on these treatments. The results of this review demonstrate the potential of the chitosan-based scaffolds for biomedical applications in cardiac tissue reconstruction as a less invasive alternative with fewer adverse side effects.

## 2. Methodology

For the extraction of the articles on the chitosan scaffolds for the treatment of myocardial infarction, a strategy based on the “preferred reporting items for systematic reviews and meta-analyses (PRISMA)” methodology was used [36].

In addition, the PICO method (problem or population, intervention, control, outcome) was used to construct the search key [37].

### 2.1. Databases and Search Strategies

For the collection and classification of the information, articles published in three databases were considered: Scopus, Pubmed and Google Scholar. The titles and abstracts of the literature published from 2017 to June 2022 related to the chitosan scaffolds used to regenerate the myocardial tissue were reviewed and identified. In Scopus, the search was performed with the following search strategy (ST): “(“cardiovascular stroke” OR “heart attack” OR “myocardial infarct” OR “infarction” OR “myocardial”) AND (“chitosan” OR “polyglucan”) AND (“tissue scaffolding” OR “tissue scaffold” OR “scaffolds”* OR “hydrogel” OR “porous scaffold” OR “cardiac patch” OR “fibrous scaffold”) AND (“cell engineering” OR “proliferation” OR “vascular engineering” OR “biocompatible”* OR “cardiac tissue engineering”)”. The terms “cardiac regeneration”, “chitosan scaffolds”, “cardiomyocyte proliferation”, “myocardial regeneration”, “gene expression”, “infarction treatments” and “cardiac scaffold” were used in Google Scholar and Pubmed in order to complement the search for information. Similarly, the filters in the databases were considered, such as for documents written in English and research documents. This review analyzed systematic reviews and research articles with information on the chitosan scaffolds and the regeneration of infarcted heart tissue in the screening stage. After identifying the studies with the search strategy, the Mendeley software automatically eliminated duplicates.

### 2.2. Eligibility Criteria

Inclusion criteria: For the inclusion of the articles that would be reviewed, the articles in English that were considered included in vivo studies with the biomodels and in vitro studies with stem cells and discussed the regenerative capacity of the chitosan scaffolds in myocardial regenerative therapy (Figure 1). Additionally, the articles with immunohistochemical tests, histology, electrocardiograms and protein analyses on the biomodels with a history of myocardial infarction treated with various chitosan scaffolds were included.

Exclusion Criteria: The information that were excluded included systematic reviews with a different scope than the chitosan scaffolds and the regeneration of infarcted heart tissue, articles with information on the scaffolds used in different fields than cardiology, epidemiological analyses, duplicates, clinical trials without using the chitosan scaffolds, book chapters and conference proceedings. In the first exclusion criterion, the selected documents from each database were loaded, and the duplicates were reviewed, discarding 84 documents (Figure 1). In the second information filter, 2098 documents focusing on water treatment, tissue engineering, and on different tissues than the heart were discarded. As the last exclusion criterion, 16 records were eliminated because they were systematic reviews without the chitosan scaffolds in vivo/in vitro or without an application in myocardial tissue regeneration.

Selection of the studies: Two researchers independently recognized and screened the titles and abstracts of the collected records by applying the criteria above. Subsequently, each study was reviewed, carefully evaluating its content to analyze the fulfillment of the eligibility criteria in a blinded way by each investigator, and the results were compared. The current information between the properties of the chitosan scaffolds and the regeneration of the infarcted tissue was delimited and selected (Figure 1).

## 3. Myocardial Infarction

MI, or a heart attack, occurs when one or more coronary vessels impede the blood flow to the myocardium, decreasing the delivery of oxygen and nutrients [2]. Various causes of MI are known, such as Kawasaki syndrome, coronary artery trauma, congenital coronary anomalies, cardiac spasms, depression, stress, hypertension, alcohol consumption, periodontal problems, diabetes mellitus, dyslipidemia, atherosclerotic platelets, platelet activation and thrombus formation [2,38]. These problems trigger cardiomyocyte necrosis, causing a tissue with low elasticity and electrical conductivity that affects the motor function of the muscle [3].

The adult heart cannot recover after an injury, and this regenerative inability of the myocardium is the leading cause of heart failure [39]. Therefore, the early detection of complications or chronic injuries decreases the probability of heart attack and death. Clinically, biochemical and instrumental tests are performed within the first six hours of the infarction to confirm the presence of the ischemic event. Some cardiac markers are used to determine a myocardial infarction through biochemical tests and monitoring aspartate, transaminase, alanine transaminase, troponin I and creatine kinase [40].

Instrumentally, muscle damage can be observed with electrocardiograms that detail the heart’s electrical activity [41]. An injury or trauma to the myocardium will affect the electrophysiology of the myocardium, with alterations in the ST-segment of the heart (the ST-segment is a flat, isoelectric line that describes the end of the depolarization, the S wave, and the beginning of the repolarization of the ventricular, the T wave) [42].

Transmural muscle damage (from the pericardium to the endocardium) due to an ischemic attack is known as ST-elevation myocardial infarction (STEMI) [42], and in non-transmural wounds, it is known as non-ST myocardial infarction (NSTEMI) [43]. Figure 2a shows the distinctive aspects of myocardial infarction, where the ventricular remodeling stage is the critical stage for heart recovery due to a decreased ventricular width.

The infarcted tissue is formed in four stages. The first is the release of cardiac biomarkers after apoptosis or necrosis of the cardiomyocytes (Figure 2, stage 1 BC death). The second stage consists of an acute inflammatory response, mediated by cellular agents, myofibroblasts, endothelial cells and macrophages that release the interleukin 6 (IL-6) and interleukin 8 (IL-8) cytokines (Figure 2, stage 2 acute inflammation). Simultaneously, macrophages help in the removal of the necrotic cardiomyocytes. In the third stage, the formation of a tissue with high collagen proteins occurs. This formed tissue provides rigidity to the muscle and a second inflammatory response by macrophages and lymphocytes (Figure 2, stage 3 Formation of fibrous tissue). Finally, the restructured infarcted tissue exhibits the formation of blood vessels. A protein-rich tissue with low elasticity and poor mechanical properties is obtained (Figure 2, stage 4 remodeling) [38,44].

Recognizing the type of heart attack facilitates the type of medical procedure required, significantly improving life expectancy. The electrocardiogram and some biochemical tests are used to describe six types of infarction: MI due to atherothrombosis (type 1), a demand–supply imbalance other than acute atherothrombosis (type 2), MI with sudden death not confirmed by electrocardiograms or biomarkers (type 3), MI due to PCI (type 4a), MI associated with a stent (type 4b) and MI related to a coronary artery bypass graft (type 5) [43,44]. Therefore, methods have been established to treat each type of myocardial infarction and reduce its clinical impact.

Conventional MI treatments mainly apply to pharmacological, reperfusion and organ transplant therapies. In the pharmacological methods, many anti-inflammatory and antiplatelet agents have been used for the treatment of MI and the prevention of a second MI. The primary drugs used include colchicine [45,46] and P2Y_12_ inhibitors such as copidrogel^®^, aspirin^®^ and rivaroxaban^®^, among others [47]. The primary function of these drugs is to inhibit the formation of endothelial platelets and inflammatory mechanisms from restoring blood flow in the heart [45,46,48]. However, many of these drugs have side effects, such as Bisoprolol^®^, a β-blocker that can increase the risk of a heart attack [49], or otamixabon^®^, which does not reduce ischemic events and increases bleeding in people with acute coronary syndromes [50]. Table 1 shows the primary drugs used and their mechanisms of action for controlling the damage caused by a heart attack.

As shown in Table 1, the conventional methods focus on preventing a heart attack or a second ischemic episode [72,73,74,75]. However, some β-blockers have been reported not to reduce CVD mortality and events after MI [76] or heart failure [77]. Additionally, the therapies that administer anticoagulants and antiplatelets have shown a risk of bleeding due to medical mismanagement, increasing the probability of hospital readmission [78,79].

On the other hand, reperfusion therapies such as CABG and PCI allow for the treatment of the damaged tissue without affecting the muscle tissue with procedures that allow for restoring the flow of nutrients to the infarcted area [67,80]. Due to the complexity of surgical procedures, they must be carried out with other therapies such as antiplatelet therapy, the use of β-blockers and, in more severe cases, heart transplantation [48,66].

Cardiac tissue engineering using biomaterials rises as an alternative therapy for MI. For example, Rosellini et al. [81] concluded that a mixture of 20:80 of an alginate:gelatin weight ratio stimulated C2C12 myoblast proliferation and differentiation. However, the authors claimed that myoblast proliferation occurred only if the gelatin was presented in the C2C12 mixture [81]. On the other hand, Algisyl-LVR™ was cited as a novel biomaterial for its beneficial therapeutic effects on the left ventricular remodeling in preclinical and clinical trials [82,83]. Secondly, tissue engineering seeks to treat cardiovascular problems using biocompatible scaffolds. The objective is to emulate the tissue’s three-dimensional porous structure to allow for the exchange of nutrients and cellular waste from the tissue revascularization damage [33]. Consequently, the scaffolds based on the biopolymers and their regenerative capacity for the infarcted myocardium have been studied. Among these scaffolds, those based on chitosan (a biocompatible, biodegradable polymer that stimulates cell regeneration) have been postulated for their application in the treatment of the infarcted tissue [84].

## 4. Scaffolds for Cardiac Tissue Regeneration

The heart’s regenerative capacity is remarkably low compared with other organs. Therefore, the primary goal of cardiac tissue engineering is to accelerate the regeneration of damaged arteries, valves, or heart muscles [85]. These results are obtained using pluripotent or multipotent stem cell differentiation processes to obtain functional mature cells [86]. Due to the rapid cell death caused by stem cells, using these cells in vivo or ex vivo studies has been considered a significant challenge in cardiac tissue engineering.

Recently, stem cells have been introduced into three-dimensional scaffolds that simulate the biological environment of the heart and supply nutrients [87]. These studies demonstrated that the differentiation and proliferation of stem cells are significantly improved if the scaffolds are prepared with biocompatible compounds, stimulating cardiac tissue regeneration [33].

Among the biocompatible compounds are natural polymers, such as alginate [88], gelatin [85] and chitosan [89], and biodegradable synthetic polymers, such as polycaprolactone [89], polypyrrole [90] and polyvinyl alcohol [91]. These polymers help renew dysfunctional tissue and provide structural support for the development of stem cells. In some cases, they are used as drug-releasing agents to enhance the regenerative processes [92]. Structurally, the scaffolds must be three-dimensional, with a good porosity for exchanging nutrients and cellular waste material with the surrounding medium [93].

The scaffolds used in tissue engineering have shown the ability to generate a microenvironment abundant in biomolecules, elasticity and a specific mechanical capacity, promoting new cell growth [94]. A scaffold applicable to the infarcted tissue should reduce myocardial scar fibrosis, induce electrical stimulation and decrease the widening of the ventricular walls [95,96,97].

A good bioactivity in the natural polymer-based scaffolds has been shown to improve the left ventricular fractional ejections in chronic heart disease [98], significantly improving the neovascularization processes in the infarcted area [99,100]. On the other hand, chitosan is a natural polysaccharide with antithrombotic and biocompatible properties that decreases the ventricular remodeling processes in infarcted hearts [100].

### Bioactivity of the Scaffolds in Myocardial Tissue Engineering

The material’s bioactivity promotes a strong bond between the scaffold and the tissue with low to negligible fibrous capsule formation [101,102].

There are two types of scaffolds capable of promoting bioactivity in the implantation areas: microfilamentous and microporous scaffolds [84], with three composition types: natural, synthetic and hybrid [89], and two forms of application: patches and injections [103]. Ideally, each must be able to conduct electricity, allow for the differentiation of the stem cells into the cardiomyocytes and have Young’s modulus similar to the native myocardial tissue [104,105]. If a scaffold meets these requirements, it can improve the cell proliferation and differentiation processes and stimulate the production of biomolecules [106].

Furthermore, promoting cardiomyocyte bioactivity is a complex process. However, it has been shown that the chitosan scaffolds can achieve this task by varying the composition of their constituents [84,103]. The chitosan-derived hydrogels encapsulate the stem cells and promote their maturation into the cardiomyocytes [107]. Chitosan, protein and stem cell hydrogels have been applied intramuscularly to promote the secretion of the growth factors that improve the myocardial conditions [108], as in the case of the expression of the vascular endothelial growth factor (VEGF), which allows for the revascularization in the infarcted area [109]. In addition, the porous and fibrous chitosan scaffolds are used as support structures to promote the stem cells’ maturation into the cardiomyocytes [110].

Similarly, it has been observed that chitosan-based scaffolds with oriented morphology can be obtained from the mixture of chitosan, gelatin and fibronectin using the 3D printing technique. This type of scaffold can induce the orientation of the muscular and cardiac tissues and implant the stem cell-derived cardiomyocytes [111]. An essential requirement to apply the 3D printing technique is moldable materials with low viscosities, allowing cellular spatial organization inside the scaffold [20]. However, pure chitosan cannot be used in the preparation of 3D scaffolds due to its high viscosity [15]. To overcome that limitation, chitosan can be functionalized due to its reactive amino and hydroxyl groups, thus obtaining chitosan derivatives with complementary properties to pristine chitosan. For example, hydroxybutyl chitosan is a derivative used by Yoshinari et a [111]. Their results demonstrated the ability of the compound to create organized tissue composed of stem cells by 3D tissue printing. Similarly, gelatin and fibronectin are simulating agents for physiological cardiac environments.

On the one hand, gelatin is used due to its high resemblance to collagen [112], while fibronectin is a binding protein in tissues throughout the body, including the heart [113]. With this, it was evidenced that the functionalization of chitosan with hydroxybutyl and the addition of fibronectin and gelatin allowed for the construction of structures with aligned pores, thus generating an oriented and united cardiac tissue with similar biological properties to the natural myocardial tissue [111]. The myocardium is characterized by its helically arranged fibers. Obtaining the oriented tissue could facilitate the distribution of the tissue elongations without the concern of a rupture due to the uniform and unidirectional distribution of the force of the heart in the diastolic and systolic cycles.

## 5. Chitosan

Chitosan (CS) is a deacetylated derivative of the second most abundant polysaccharide, chitin [114]. Chitin can be obtained through chemical or enzymatic extraction from sources such as the shells of crustaceans, insects and the mycelium of filamentous fungi [32]. The final result of the chemical extraction of chitin is a polymeric structure composed of the units of β-(1,4) D-glucosamine and N-acetyl-D-glucosamine (Figure 3A) that are soluble in dilute acid solutions (pKa 6.3) and are converted into a cationic polyamine [15]. When the degree of deacetylation (DD) of the chitin is between 50% and 99%, it is considered chitosan. It is essential in any research using chitosan to determine the degree of deacetylation and molecular weight since those are the essential factors influencing the biological response of chitosan [115].

Wang et al. [116] reported that acylation occurs mainly with acylating agents such as organic acids and derivatives, including the acylated aliphatic and aromatic groups. The esterification reaction of chitosan occurs between the hydroxyl groups of chitosan and the carboxyl groups of the organic acids. These reactions are catalyzed with inorganic acids, such as CH_3_SO_3_/P_2_O_5_ and H_2_SO_4_, or can occur through Steglich-type esterification conditions [117]. Substitutions of the hydroxyl and amine groups using etherification reactions and alkylating agents have also been reported. The most used reactions for the N-alkylation or O-alkylation of chitosan are carried out with halogenated alkanes, epoxides, or aldehydes [116]. Figure 3 schematically shows some of the chitosan functionalization reactions reported in the literature. The functionalization of chitosan has been used to improve its biocompatibility, solubility, mechanical resistance, antimicrobial action, anticoagulant properties and blood compatibility [118,119,120].

**Figure 3 molecules-28-01920-f003:**
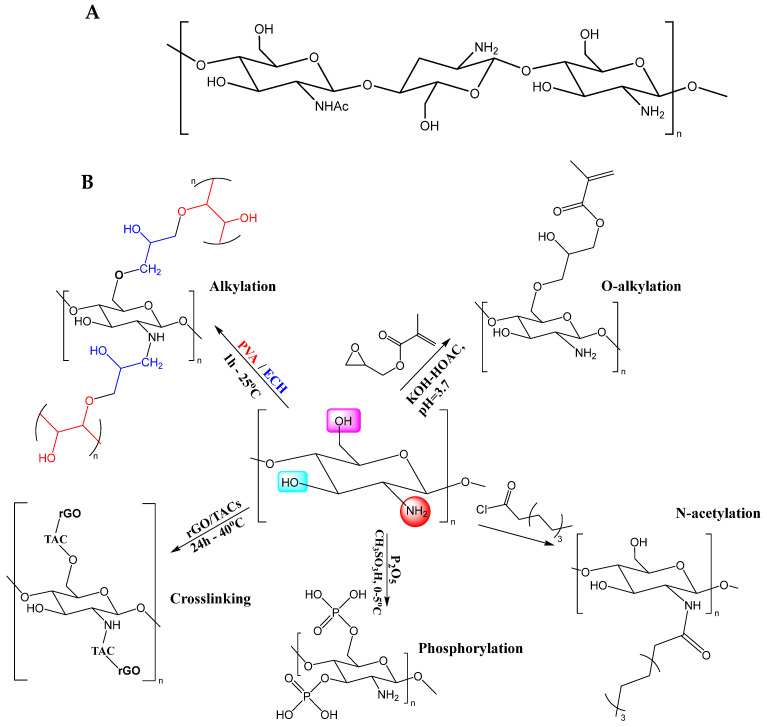
(**A**) Chemical structure of chitosan. (**B**) Synthetic methods commonly used for functionalizing chitosan in tissue engineering: alkylation [121], O-alkylation [115], N-acetylation [115], phosphorylation [122] and crosslinking [123]. Reactive functional groups of chitosan: Magenta box, hydroxyl at C-6. Cyan box, C-3 hydroxyl. Red circle, primary amine at C-2. Abbreviations: rGO = reduced graphene oxide, TAC = tannic acid, PVA = polyvinyl alcohol, ECH = epichlorohydrin, HOAc = acetic acid.

Chitosan has received good attention in tissue engineering applications due to its easy use in the three-dimensional porous structural preparations, which is essential for tissue regeneration since it confers the stability and biodegradation to the material [114].

One of the main problems of the scaffolds in regenerating the skin tissue or complex tissues, such as nerve or muscle tissue, is avoiding aggressive immune responses in the implantation area [124]. However, combining biomaterials such as chitosan and some other biopolymers can promote the biocompatibility with almost a negligible immune response in the implantation site, while antibacterial and other beneficial properties can be introduced. For example, Yang et al. [125] prepared injectable, self-healing, biocompatible and antibacterial hydrogels based on ε-poly(l-lysine)-modified poly(vinyl alcohol) (CPVA-g-EPL)/chitosan (CS)/Ag complexes using oxidized dextran as a crosslinker to promote wound repair. The hydrogels were formed through an imine dynamic Schiff base linkage between the amine groups of EPL and CS and the aldehyde groups of ODEX, which are responsible for the self-healing properties. The hydrogels presented a macroporous interconnected structure with 40% degradation after seven days of immersion in a PBS solution. However, the chitosan amount in the hydrogels was essential to retain more weight due to a higher crosslinking ratio. Remarkably, the in vitro antibacterial properties exhibited by the hydrogels against *Escherichia coli* and *Staphylococcus aureus* increased with the increasing chitosan and EPL grafting ratio and further enhanced the introduction of the silver nanoparticles. The proposed antibacterial mechanism was the cell wall and the membrane disruption revealed by the green fluorescent STYO 9 test results. According to the histology results, the CPVA-g-EPL/CS hydrogels promoted the wound healing process in a mouse skin defect model and significantly accelerated the re-epithelialization of the damaged tissues within eight days. However, introducing the silver nanoparticles did not accelerate the wound healing process. Finally, the hydrogels exhibited an acceptable low cytotoxicity without compromising the wound dressing application [125].

Chitosan with human endothelial stem cells and platelet-enriched plasma was applied to elaborate a scaffold with a porous morphology. Its role in the reconstructive engineering of nervous tissue demonstrated that the endothelial cell growth and cell regeneration were observed in the spinal cord without forming a fibrous tissue [93]. On the other hand, Castro et al. applied chitosan to develop membranes for subdermal implantation studies [126]. The authors claimed that the addition of 1.0 wt. % and 1.5 wt. % of tea tree essential oil (TTEO) to the chitosan (CS)/polyvinyl alcohol (PVA) membranes increased the biodegradation in the rats’ subcutaneous tissue. A good absorption of the membranes in the rats at 90 days of implantation was observed without an aggressive immune response and with an evident recovery of the cellular architecture. However, one aspect that requires more research is that the study concluded that the membranes with a higher content of TTEO “showed more significant signs of reabsorption, which indicates that the incorporation of the oil promotes greater reabsorption and biocompatibility.” This statement is only valid for the subcutaneous tissue and not in general, so more in-depth studies are required for such a conclusion [126].

These investigations demonstrated that some chitosan-derived scaffolds have a preliminary biocompatibility in the tissues evaluated and might promote their use in cardiac tissues. Therefore, some common findings with those investigations were the adherence and proliferation of the cardiomyocytes and the presence of gene encoding proteins in the implantation zone of the scaffold [84].

### 5.1. Chitosan’s Role in Cardiovascular Regenerative Therapy

Due to the low regenerative capacity of the cardiac tissue in mammals, the heart cannot regenerate after an ischemic episode [39]. Therefore, the chitosan scaffolds, including natural and synthetic biopolymers, have been applied to optimize the material’s mechanical properties [100]. The scaffolds that mimic the heart’s extracellular matrix prevent adverse processes in the infarcted cardiac tissue [100]. However, one aspect that requires attention and is the driving force for future research is that the chitosan scaffolds’ stimulatory mechanism within the heart is unknown. Studying the chemical, biological and morphological interactions in detail could help clarify this biomaterial’s interaction mechanisms before preclinical trials, which would increase the application of chitosan in cardiac regenerative medicine [100].

Regenerative cardiac therapy has proliferated due to the synthesis based on the scaffolds designed using biopolymers. It has been shown that the properties of some natural polymers allow the cellular cardiomyoplasty processes to be carried out in situ [127]. The approach allows for stem cells to be implanted into the infarcted myocardium or the edge of the infarcted area in order to improve the ejection fractions. However, some clinical trials have shown that only 10% of some stem cells proliferate since the rest of the cells die due to apoptosis, inflammatory responses and poor anchorage in the tissue, among other causes [128]. The scaffolds with myocardial-like properties address these problems. Mainly, the scaffolds will allow for the diffusion of oxygen and nutrients to the cells, the components that decrease significantly after the infarction. Similarly, the values of less than 40% of the cells in apoptosis are obtained after implantation [129]. For this reason, researchers use peptides, proteins and components with a high biocompatibility to avoid an immunoaggressive response on the implanted cells, ensuring the survival of the stem cells in cellular cardiomyoplasty [128].

In order to study cell evolution in the cardiac cell implantation processes, natural chitosan-based scaffolds have been used alongside other naturally occurring polysaccharides, proteins, biomolecules and polymers [101]. Using the natural compounds with chitosan will decrease the cytotoxic responses due to improved biological ligand–receptor interactions in the body [130]. In addition, these scaffolds simulate the physical and biochemical properties of the myocardium, improving the stem cell differentiation processes, migration, retention and proliferation of the native cells in the internal structure of the scaffold [107]. It is essential to emphasize the need for detailed studies into the properties of the scaffolds, such as the gelatinization time, absorption radius, morphology, rheology, structural stability in the physiological medium, cytotoxicity and behavior in the model tissues of the scaffolds based on biopolymers, such as chitosan, since these properties influence the bioactivity of the scaffolds.

For example, it has been shown that protein-containing hydrogels can promote cell growth and anchoring by promoting the maturation of the stem cells within the hydrogel [108]. However, the rapid biodegradability of the protein-containing hydrogels, such as gelatin hydrogels, limits the time required for progenitor cell maturation [108,131]. Thus, incorporating chitosan with proteins such as gelatin or collagen to create hydrogels prolongs the life of the hydrogel, improving the maturation of the cardiomyocytes, mimicking the extracellular matrix of the myocardium and supporting the restoration of the cardiac function [132,133].

A widely studied property of the scaffolds is the pore diameter. It has been reported that pore sizes between 60–200 μm generated good cell–cell communication, growth, anchoring and nutrient transport to the cardiomyocytes and cells, increasing the scaffolds’ in vitro degradability [134,135,136].

The incorporation of some bioactive molecules found in the extracellular matrix, such as hyaluronic acid, whose primary function is cell anchoring, connective tissue formation and the differentiation of stem cells into endothelial and cardiomyocytes [137], has allowed for the preparation of scaffolds with chitosan to repair heart defects in infarcted rats. The results of single photon emission computed tomography demonstrate that these scaffolds can maintain healthy post-infarction tissue with a debatable regenerative capacity [107]. The material can be applied directly to the myocardium through intramuscular injections, and its regenerative property can be increased using mesenchymal stromal cells [107]. However, the scaffold without mesenchymal stromal cells in the infarcted tissue does not present significant improvements in the hemodynamic and regenerative parameters due to its lack of biological activity in the heart.

Historically, collagen and the extracellular matrix have been used in cardiac regenerative therapy due to their natural presence in the heart [106,138]. As they are both of protein origin, they are susceptible to denaturation and a possible decrease in the communication capacity and cellular anchorage of the stem or cardiac cells in tissue regeneration therapies [138,139]. However, the preparation and application of the scaffolds containing collagen and the extracellular matrix using the techniques of sample heating, such as electrospinning in the infarcted cardiac tissue, are limited due to their possible protein denaturation, which significantly affects the biological properties of collagen [140]. Chitosan is used to develop some scaffolds through crosslinking the reactions to avoid denaturation and maintain the biological properties of collagen and the extracellular matrix [139,141]. Crosslinking collagen with chitosan also reduces the percentage of denaturation up to 30% when preparing the scaffolds using the electrospinning [141] or lyophilization [132] techniques. The in vitro assays allowed for the design of the scaffolds with a controlled macroporosity (25–200 μm) and a good cell viability of the cardiomyocytes from the neonatal rats (above 70%) [132,141]. Similarly, mixing chitosan with the extracellular matrix helped modify the scaffold’s porosity by significantly maintaining and promoting the fluid absorption, cell anchorage and degradation in vitro [139].

On the other hand, Tsukamoto et al. [111] designed heart tissue using 3D printing techniques. The methodology used gelatin and fibronectin to simulate the heart’s biological and protein structure. However, hydroxybutyl-functionalized chitosan gave the scaffold a better malleability and cardiac tissue with the required properties, including tissues based on the aligned cardiomyocytes [111].

The role of chitosan in the cardiac processes of rats was established in the research by Efraim et al. [106], in which it was shown that chitosan improved the mechanical properties of the porcine decellularized extracellular matrix and maintained stable heart hemodynamic parameters in the rats with chronic myocardial infarction [106]. These results may be linked to the segregation of the genes in the infarcted area, stabilizing the tissue and the hemodynamic parameters of the heart [106].

#### 5.1.1. In Vitro Assays of the Chitosan Scaffolds in Cardiovascular Regenerative Therapy

It is possible to obtain chitosan-based hydrogels with beneficial antioxidant properties for cell proliferation in oxidative biological environments, such as in the infarcted myocardium [132]. After infarction, the reactive oxygen species (ROS) content rises in the myocardium, causing apoptosis, fibrillation and muscle damage [132]. Gao et al. [142] showed that the chitosan hydrogels incorporating vitamin C increased the cardiomyocyte survival in a simulated oxidative stress environment. This mechanism can be explained by the generation of free radicals from the ascorbic acid (AA), which donates electrons to the ROS, forming the AAH• complex, followed by the extraction of hydrogen from the hydroxyl and amino groups of chitosan. These results suggest that using antioxidant agents with chitosan synergistically improves the antioxidant action of the hydrogels, preventing the cells seeded in the infarcted myocardium from being affected by oxidative stress environments.

On the other hand, chitosan mixtures with some synthetic polymers or nanomaterials can conduct electricity and expand its ability to restore the properties affected by myocardial infarctions to the heart [130,132]. Hybridizing the chitosan scaffolds enhances the electrical propagation in the infarcted area, allowing electrical communication between the functional native tissue, necrotic tissue and stem cells implanted in the heart, triggering the synchronization of the diastolic–systolic cycles [129,130,143]. Telabi and colleagues [103] succeeded in synthesizing a patch-like scaffold. Their results showed that by adding 7.5% of polypyrrole to a mixture composed of chitosan-polycaprolactone (CS-PCL) (CS:PCL 7:3), a significant increase in the electrical conductivity was evidenced. Additionally, the obtained scaffold presented bactericidal properties against *E. coli* and *S. aureus* [103]. Despite the promising results by Telabi and co-authors, a detailed scaffold compatibility analysis demonstrating the biosafety of the scaffolds prepared with the cardiac tissue cells has not been reported.

Similarly, Kalishwaralal et al. [144] reported on a chitosan scaffold with selenium nanoparticles. This scaffold presented antioxidant and electroconductive capacities of 0.0055S cm^—1^. The Kalishwaralal and Telabi results demonstrate that the chitosan-based scaffolds with conductive polymers or nanoparticles have the potential to enhance the electrical conduction in the cardiac tissue, although this aspect must be tested in situ directly in infarcted tissue. However, it is pertinent to further investigate the cytotoxic capacity of the materials used to improve the electrical conductivity. This information would allow for the results to be scaled up for preclinical and clinical trials with a greater biosecurity.

Chitosan hydrogels with the protein components as the extracellular matrix can help to improve the specific functions in the cardiomyocytes. Tsukamoto et al. [133] designed a thermosensitive chitosan hydrogel functionalized with 1,2-butylene oxide and, in turn, mixed with fibronectin and gelatin. According to their results, the hydrogel can be used to create heart tissue using 3D printing techniques. Tsukamoto and colleagues cited that the tissue was elaborated due to the fact that the functionalized chitosan provided the structural support, and the mixture of fibronectin with gelatin induced the anchoring and the cell attachment of the tissue [133].

The group, in 2020, produced a vascularized cardiac tissue for cardiac co-therapeutic purposes [111]. Tsukamoto and co-authors used the same methodology in this study to develop the thermosensitive chitosan hydrogel. Their results demonstrated that hydroxybutyl chitosan with some protein components helps create a cardiac tissue compatible with the cardiac epithelial cells and cardiomyocytes. Similarly, Goldfracht et al. [145] studied gene segregation in a tissue imprinted with the chitosan scaffolds and the extracellular matrix. Their results showed that the cardiomyocytes in the chitosan-based structures can carry out the processes to achieve cell maturation. These researchers demonstrated that the structures based on chitosan could significantly improve the cellular properties of the cardiomyocytes or some specific cell groups. Using these tissues in regenerative therapies for the infarcted myocardium could present good results due to the scaffold’s composition and tissue formation using printing techniques, such as layer by layer (LBL). Consequently, applying chitosan and other biomaterials for the scaffold’s preparation has shown indications of potential therapeutic properties, such as the antioxidant, regenerative, electroconductive and thermosensitive properties.

Ke and colleagues succeeded in designing an injectable hydrogel with stem cells. The results showed that using 1.77% chitosan contributed to maintaining the cellular and structural morphology of the post-application hydrogel [146]. They demonstrated that, due to the amino groups easily protonated in an acidic medium, chitosan can interact electrostatically with the other biopolymers and molecules, such as dextran and β-glycerophosphate, and trigger the properties such as the degradation time, viability, cell release and diameter pore [146]. By controlling the pore diameter, better degradation results, the encapsulation of the mesenchymal cells and a decrease in the apoptosis of the umbilical cord mesenchymal stem cells (UCMSCs) occur. There was a release of the extracellular signal-regulated phosphorylated protein kinase 1/2 complex and phosphorylated protein kinase B (*p*-ERK1/2 and *p*-Akt, respectively), indicating that controlling the structural properties of chitosan promotes the reconstructive activity. In addition, they established that the scaffold could promote cell survival and retention, aiding the differentiation of the specific stem cells. However, it is necessary to know the role of chitosan in the protein segregation mechanisms for the reconstruction of the infarcted tissue in vivo, since its regenerative capacity was not evaluated in the biomodels [146,147]. Similarly, decreasing the apoptosis of the umbilical cord-derived mesenchymal stem cells (UCMSCs) reinforces the theory that the natural chitosan scaffolds could play a key role in restoring the infarcted myocardium and maintaining the cardiac functions in the short term.

Some chitosan scaffolds mixed with the proteins belonging to the myocardium can prevent the apoptosis of some cells and reconstruct the infarcted tissue, maintaining some hemodynamic parameters, such as the internal diameter of the left ventricle in the diastole and systole and the thickness of the posterior wall of the left ventricle in the diastole and systole, as well as somefunctional parameters, such as the fractional area change, ejection fraction and fractional shortening using echocardiographic techniques [106,148]. Currently, the role of the chitosan materials for improving the hemodynamic parameters of the heart is not discussed in-depth when the natural hydrogels with chitosan are applied. Many authors focus the attention of their studies on the other components of the scaffolds, ignoring the biological contribution of chitosan. More blotting studies should be performed to fully understand the ability of chitosan to promote and maintain the hemodynamic parameters of the heart.

#### 5.1.2. In Vivo Assays of the Chitosan Scaffolds in Cardiovascular Regenerative Therapy

Two types of minimally invasive surgical techniques are applied in the in vivo tissue engineering trials for MI: regenerative epicardial patches and injectable intramyocardial hydrogels. The cardiac patches involve the release of the bioactive chitosan stimulate growth factors [149]. The injectable hydrogels, on the other hand, are used to encapsulate and release the cells and drugs in a controlled manner to generate angiogenesis and decrease the pathological ventricular remodeling [149].

Pok et al. [150] designed a three-dimensional composite cardiac biodegradable patch from self-assembled PCL (mixture of 10 kDa and 80 kDa to form a microdomain-like indentations that permitted the chitosan–gelatin hydrogels) sandwiched in a gelatin–chitosan hydrogel to reconstruct the congenital heart defects. The electron microscopy analysis determined that a porous structure with a mean pore diameter of ~80 µm and the migration of the neonatal rat ventricular myocytes (NRVM) showed a cell viability for seven days. The prepared multi-layered scaffolds exhibited a tensile strength and degradation rate dependent on the PCL molecular weight, with PCL lower molecular weights showing lower tensile strengths and faster degradation rates, but sufficient support for a full-thickness ventricular defect. The scaffolds showed a similar compressive modulus (~15 kPa using 50% gelatin: 50% chitosan) to the native cardiac tissue. The authors highlighted that the hydrogels with the presence of 50% gelatin improved the homogeneity and geometry of the pores and the interconnectivity, while the scaffolds prepared with an excess of gelatin (75%) or PCL alone decreased the cell adhesion.

It is essential to note that the pure chitosan scaffolds could not support the cardiomyocytes’ adhesion and proliferation, despite their controlled porosity [151]. In contrast, the gelatin scaffolds did not have enough tensile strength and stability to be used as the cardiac patches despite their biosorption, biocompatibility and capacity to maintain the cell viability [152]. However, the chitosan and gelatin scaffolds were structurally stable in the cell culture media [153]. For instance, the correct combination of the 10 kDa + 80 kDa PCL thin layer core that offered mechanical surgical support and a tensile strength with the 50% gelatin+50% chitosan porous hydrogel that provided cell adhesion, migration and maturation were responsible for the improved in vitro results. They also replaced gelatin with a decellularized porcine heart extracellular matrix (ECM) in the multi-layered scaffolds, obtaining an improved contractile and electrophysiological functions in a neonatal rat ventricular myocyte model [154].

The same group, in 2017, reported that, using a rat model of the right free wall replacement, the transplantation of the previous PCL core and chitosan/gelatin or the chitosan/heart ECM produced significant muscular and vascular remodeling and resulted in a significantly higher right ventricular ejection fraction compared to the use of a commercially available pericardium patch after eight weeks of implantation [155]. This was primarily attributed to the endothelial cell infiltration and the invasion of the engineered multi-layered scaffolds presented in contradistinction to the commercially available pericardium patch, as evidenced by the hematoxylin–eosin (H&E) staining and the quantitative muscular and endothelial cell analysis that reported a higher cell density [155]. In addition, the heart extracellular matrix patches produced a denser vascular network and active participation of the macrophages in the remodeling process, as evidenced by the immunofluorescence staining that exhibited the presence of the expression markers CD86 (M1) and CD206 (M2). Interestingly, the number for both the markers increased in the area of the patch implantation after four weeks without significant differences between the gelatin/chitosan or the heart ECM/chitosan patches. However, the heart ECM patches exhibited higher M2/M1 ratios, indicating a faster constructing remodeling ability than the gelatin patches due to the higher M2 macrophage presence.

Additionally, using stem cells or stromal mesenchymal cells in regenerative medicine with the chitosan scaffolds is common. These cell groups induce the reconstruction of the damaged tissues and the replacement of specific tissues, but the misuse of these cells could generate detrimental damage to the patient models or compromise the safety of the assay [156]. Rabbani et al. [107] used Wharton jelly stem cells, indicating that angiogenesis and the possible formation of cardiac tissues were evidenced.

Consequently, the cardiac functions in the small models (mice) remained stable [107]. However, the differentiation to the cardiomyocytes was not possible because the Wharton jelly stem cells differentiate, for the most part, into endothelial cells by releasing the CD31 marker [157]. It was evident that the preservation of the tissue and hemodynamic functions in the mice was caused by the high levels of the cardiac marker MHC-β and not by the segregation of the cardiac marker CD31, as discussed by Rabbani and colleagues [107]. On the other hand, it was possible to obtain the stem cell differentiation when they were encapsulated in a 1.77% chitosan hydrogel, 1.0% Dextran and using β-glycerophosphate as a crosslinking agent [146].

On the other hand, Saravanan et al. [136] synthesized a biodegradable and electroconductive scaffold from clinical-grade chitosan with Au–graphene oxide nanosheets to treat the rats with myocardial infarction. The authors documented the inverse relationship between the pore size and the chitosan concentration [136]. The electrocardiograms of the infarcted biomodels showed a significant improvement in the intervals of the QRS complex due to an increased electrical conductivity and contractility in the area of the infarction, accompanied by an upregulation of the connexin 43 gene within five weeks. These findings note that chitosan can interact electrostatically with the other constituents of the scaffold, allowing for the formation of pores with smaller diameters and hindering the enzymatic hydrolytic degradation on the glycosidic bonds of chitosan.

Additionally, the scaffold showed good electrical conductivity when 0.5% GO–Au and 2% clinical-grade chitosan were used, efficiently improving the tissue contractility. In this context, Savaranan et al. [136] argued that improving the electrical propagation of the scaffold will keep the intervals of the QRS complex of the electrocardiogram stable, indicating a statistically similar cardiac conduction and contraction velocity concerning the control [136]. Both results showed that modifying the chitosan concentration allows for controlling the scaffold’s porosity and interaction with the surroundings molecules and increases the cell proliferation through the improved cell communication and nutrient uptake from the extracellular matrix [136,139].

In another study, He et al. [158] indicated that chitosan was not capable of conducting or propagating electrical voltage when applied to mice by itself. However, some nanostructures helped to obtain the scaffolds with electroconductive properties, mainly when carbon nanostructures such as graphene oxide were used, e.g., MWCNTs and SWCNTs [84]. Additionally, chitosan mixed with synthetic polymers such as polylactic acid (PLA) [159] or polycaprolactone (PCL) [160] present synergy, helping to diffuse the electrical potentials in the infarcted zone and promoting the synchronization of the cells inside, the segregation of the proteins and the cell–cell and cell–environment communication [158].

In MI regenerative therapy, stem cells also play an essential role in regenerating infarcted tissue. Mesenchymal stem cells are adult, pluripotent cells widely used in regenerative therapy and tissue engineering. Bin X. et al. [161] studied the capacities of the injectable hydrogel composed of chitosan and β-glycerophosphate to restore the cardiac structural functions. Their results demonstrated that the thermosensitive hydrogel allowed for the differentiation of the mesenchymal stem cells into myocytes. In addition, the hemodynamic heart parameters were normalized for the control, and indications of revascularization in the infarcted tissue were reported [161]. However, Yang L. et al. [148] demonstrated that the Caspase-1 hormone levels were decreased in the chitosan, β-glycerophosphate and mesenchymal stem cell hydrogels. The decrease in Caspase-1 was related to endothelial cell pyroptosis, thus explaining the evident formation of the tissue [148]. Both studies demonstrated that the chitosan scaffolds with the mesenchymal stem cells increased the capillary density in mice, but both failed to explain in detail how the chitosan scaffolds loaded with the stem cells were able to normalize the cardiac hemodynamics [148].

These results demonstrate the versatility of the chitosan hydrogels for regenerative cardiac therapy. Nevertheless, it has been shown that some nanomaterials, such as graphene oxide and TiO_2_, in high concentrations in the chitosan scaffolds negatively affect the cells, producing the apoptosis processes in specific stem cells [162,163]. Nanomaterials, such as graphene, gold and selenium nanoparticles, have been used to mimic and synchronize the systolic–diastolic cycles of the cardiomyocytes in the porous scaffolds, significantly improving the electrical conductivity [135,136,144], the control of the structure and the mechanical properties [164,165].

Recently, a hydrogel based on chitosan, nanobioglass and γ-glutamic acid was reported as a promising hydrogel, promoting the revascularization of the infarcted zone with the help of the vascular endothelial growth factor (VEGF) that is secreted in the zone [166]. Finding the VEGF in the myocardium is closely linked to the revascularization processes, with the possible hypertrophy or differentiation of the exogenous seeded-cell in regenerative therapies [167].

Finally, more studies are required to analyze the in vitro and in vivo cytotoxicity of hydrogels, their long-term stimulating capacity, the necessary balance in their composition that promotes better properties without adverse side effects and the preclinical studies that demonstrate their potential application directly for MI in humans.

In the case of the scaffolds with fibrous characteristics, they cannot be directly injected into the myocardium or encapsulate the cells inside, as in the case of hydrogels. Instead, the scaffolds with a fibrous structure fulfill the task of providing structural and stimulatory support to the stem cells in regenerative therapy [168]. However, as with the hydrogels, producing fibrous scaffolds exclusively from chitosan has some drawbacks, such as a low capacity to form fibers using the electrospinning technique and a low electrical conductivity [169]. This problem has been solved by combining chitosan with other polymers with better mechanical properties, such as polyvinyl alcohol (PVA) [170].

Ahmadi et al. [168] demonstrated that it was possible to elaborate using electrospinning, a fibrous scaffold composed of chitosan, polyurethane and carbon nanotubes. The authors reported that microfilaments could be obtained if 2 wt. % of chitosan was functionalized with the carboxyl groups of the carbon nanotubes. Their results established that functionalizing the carbon nanotubes with chitosan promoted hydrophilicity, roughness, biocompatibility and cell proliferation [169].

The advantages and disadvantages of the studies related to the chitosan scaffolds for the treatment of MI are observed in Table 2.

The chitosan scaffolds are mainly studied to support stem cells and promote their differentiation to the cardiomyocytes with electrical stimulation, endorsing their application in cardiac regenerative therapy. Unlike the scaffolds prepared with the other biopolymers, such as polylactic acid, the chitosan scaffolds considerably improve the hydrophilicity of the material. Generally, the hydrophilicity is determined by two properties, the pore size and the constitution of the generated fibers or pores [160]. The proper hydrophilicity and pore size stimulate the cell proliferation, nutrient exchange and communication [168]. Adding chitosan to the polyurethane scaffolds improved the biocompatibility and hydrophilicity, making it possible to obtain fibers with optimal diameters for the cell adhesion and proliferation [168,175].

## 6. Role of the Chitosan Scaffolds in Myocardial Tissue Regeneration: Stimulation of the VEGF Growth Factor Secretion

It is essential to highlight that the malleability of chitosan is critical for obtaining the organized filamentous structures that improve communication between the cells. The high organization obtained in the chitosan patches allows for an improved exchange of nutrients and waste material and a better contractile capacity of the tissue due to the optimal distribution of the mechanical load [163]. On the other hand, chitosan has high biocompatibility with cardiac tissue, and its production is relatively simple, expanding its application potential. Similarly, its chemical versatility expands the possibilities of its chemical functionalization, allowing for an extensive repertoire of materials with diverse physical and chemical properties to be applied in cardiac tissue engineering [115,120,176].

Chitosan functionalization also facilitates the incorporation of the scaffolds in the body and the interaction with the extracellular matrix [177]. Several experiments have observed that collagen type I secretion supported the cell adherence processes when the patches loaded with the stem cells of the stem cell-derived cardiomyocytes were used to replace the necrotic cardiomyocytes. For instance, using the conductive patches loaded with mesenchymal cells can improve the restoration of the hemodynamic and biomechanical functions [134,178,179].

Electrically stimulating the native or seeded cardiomyocytes in the scaffold also generates the upregulation of the growth factors and proteins required in the stem cell differentiation, maturation and proliferation. The overexpression of some biomolecules is due to two possible factors: the mechanical stimulation and the electrical stimulation of the receptors, such as the vascular endothelial growth hormone receptors (VEGFR1 and VEGFR2), a hormone involved in the angiogenesis processes in various tissues [180].

Recently, it was determined that the chitosan scaffolds could stimulate the VEGF family’s glycoprotein receptors, specifically VEGF-A [166]. The homologous receptors of VEGF-A, VEGFR1 and VEGFR2 are found in the cell wall of the cardiomyocytes, whose primary function is to promote angiogenesis in the myocardial tissue [180]. VEGF-A is released following the myocardial tissue injury or inflammation by subjecting the cardiac tissue to mechanical stimulation or through contact with the transforming growth factor-β (TGF-β), which improves the migration of stem cells and endothelial cells on the infarcted tissue and the survival of the cardiomyocytes, producing greater permeability and contractility in the tissue. Figure 4 shows a possible mechanism where VEGF-A is secreted through stimulation by the scaffolds.

It is well recognized that finding the VEGF in the myocardium is closely linked to the revascularization processes with possible hypertrophic or dividing effects of the cardiomyocytes in regenerative therapies [167]. In addition, the inhibition of apoptosis in the cardiomyocytes and a good bioactivity and biocompatibility were demonstrated [166]. The VEGF-A function was also observed when Wang et al. [181] stimulated the cell wall of the cardiomyocytes by applying a nanofiber cardiac patch composed of chitosan and calcium silicate [181]. This study observed that the chitosan and silicate ions managed to increase the expression of the VEGF, reducing the infarcted area and improving the heart’s mechanical properties and stem cells’ maturation.

## 7. Concluding Remarks and Perspectives

In recent decades, advances in the infarcted myocardial tissue engineering have aroused interest among researchers from various science disciplines and in biomedical applications. The synthesis of chitosan scaffolds with proteins and with natural and synthetic biopolymers incorporating stem cells was tested in order to improve the physiological and biochemical conditions of the infarcted myocardium, with promising results due to the implementation of the electrical conductivity, stimulation of the cell differentiation to the cardiomyocytes and improvement of the hemodynamic capacity.

In this systematic review, we discussed the role of chitosan scaffolds in the regeneration of infarcted heart tissue, analyzing their regenerative capacity and the ability to stimulate the proliferation and differentiation of stem cells into functional cardiomyocytes, the antioxidant capacity in the environments of oxidative stress and an expression of growth factors for tissue regeneration. The literature reviewed claimed that the revascularizing property of chitosan stimulates the recovery of the tissue damaged by the blockage of one or more vital arteries of the heart, demonstrating that chitosan might improve the treatment of ischemia.

Chitosan scaffolds have been used extensively for the transport and maturation of stem cells and induced pluripotent stem cells. Each scaffold developed a unique therapy with different results in revascularization and scar tissue removal through regenerative action, among other significant findings. Regarding safety, the preclinical tests in small animals showed an acceptable biocompatibility with promising results for improving some cardiac parameters, without neglecting that some materials did not show regenerative activity when used in vivo. Chitosan scaffolds are advancing in their efficacy. However, they must be introduced with stem cells to improve the infarcted tissue regeneration ability. Using growth factors would improve the tissue regeneration of these biomaterials.

On the other hand, the use of chitosan in myocardial regenerative therapy has not been fully explored. There is still much information to be analyzed on these biomaterials and, thus, more studies are required. It is perilous to conclude whether or not chitosan is efficient in the regeneration of the infarcted myocardium due to the low volume of published research. More research is required to demonstrate the restorative power of tissue regeneration from the chitosan-based scaffolds incorporated with stem cells.

The cellular regeneration of the cardiomyocyte with this type of structure is not yet fully understood, nor is its biochemical mechanism. This phenomenon may be due to the tissue complexity and the ethical concerns about the biomodels. Many in vivo studies were carried out through lateral thoracotomy, representing a high risk to the animals due to the invasiveness of the technique. Alternatively, the injectable chitosan scaffolds, including the growth factors, are possible candidates for restoring infarcted tissue in the myocardium due to their observed regenerative, therapeutic and minimally invasive factors. These injectable scaffolds facilitate the restoration of cardiac tissue since they mimic the cardiac biological environment by including biomolecules from the ECM and promoting the diffusion of the electrical potentials, attenuating the immune response on the scaffolds. These properties prolong the cell survival and increase the rate of maturation of the stem cells to the cardiomyocytes.

A highly relevant aspect that was observed in the literature review was that, in addition to the low number of publications on chitosan scaffolds in the regeneration of infarcted tissue, many articles discuss the regenerative mechanisms of the chitosan scaffold in a very general way. It is not chemically specified how chitosan contributes to tissue regeneration. We have proposed a mechanism in which chitosan interacts with the cardiac environment when applied to the infarcted myocardium of some animals. To confirm this mechanism, more studies are required on the design of the porosity and its influence on the mechanical and electrical properties and the cellular gene response around these structures. In the future, the improvement of this technology for incorporating stem cells and the regenerative stimulation of the myocardium is expected, offering a friendlier and less invasive alternative to therapy for patients. In addition, more preclinical studies are expected to explore the full potential of the scaffolds based on this natural biopolymer with revascularizing properties.

## Figures and Tables

**Figure 1 molecules-28-01920-f001:**
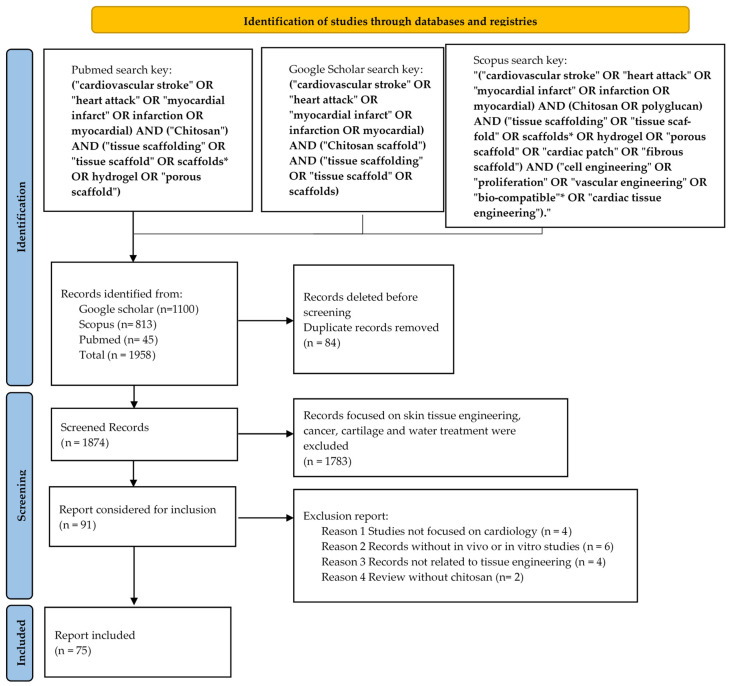
PRISMA 2020 methodology [36] was used to search for the information.

**Figure 2 molecules-28-01920-f002:**
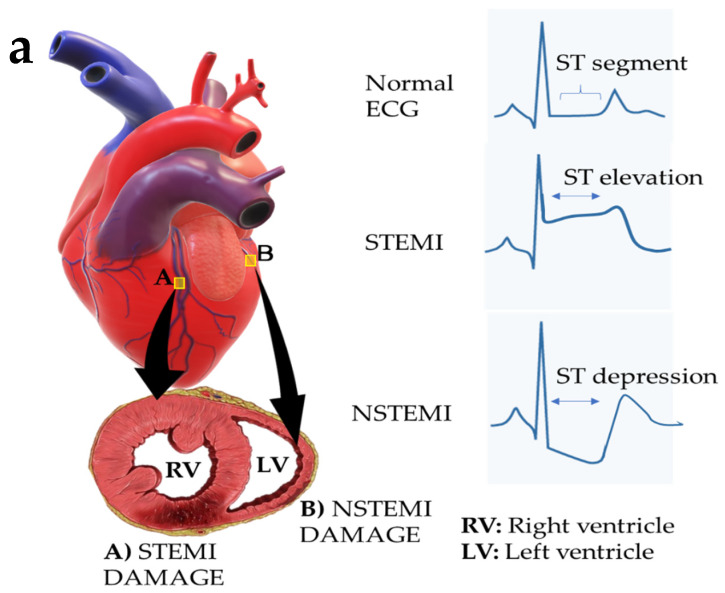
Graphic representation of myocardial infarction; (**a**) non-transmural and transmural myocardial infarction, (**b**) the formation of the infarcted tissue in the heart after an ischemic process. (1) Cell death in the first hours of infarction. (2) Cytokine release and degradation of the necrotic cells by macrophages. (3) Muscle fibrosis processes due to a high protein content in the infarcted area and dysfunctional cell degradation. (4) Final fabric. Abbreviations: ECM, extracellular matrix; BMs, biomarkers; CMs, cardiomyocytes; NSTEMI, non-T-segment elevation myocardial infarction; STEMI, T-segment elevation myocardial infarction; EGC, electrocardiogram [38,44].

**Figure 4 molecules-28-01920-f004:**
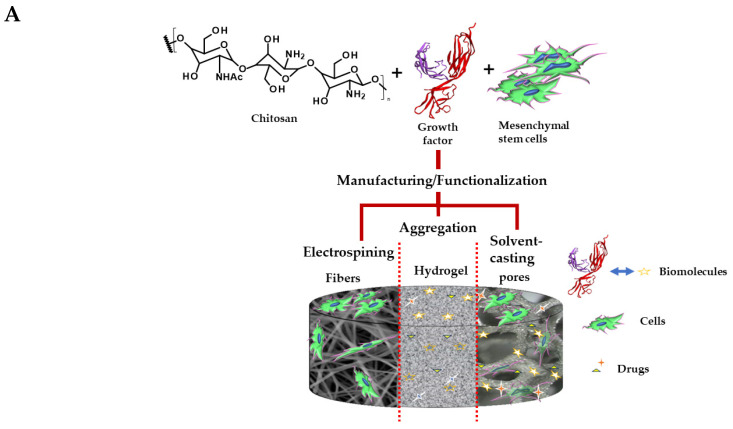
(**A**) Elaboration of the scaffold, constituents and uses. (**B**) Possible mechanism of cellular regeneration and differentiation of the cardiomyocytes. The chitosan scaffold allows for the transfer of exogenous cells to the damaged tissue. In the process, the scaffold surface mechanically stimulates the VEGFRs. Simultaneously, the VEGFR can be electrically stimulated by natural electrical stimuli from the heart or external sources due to the functionalization of chitosan with the other electro-conductive polymers (represented in the figure as R). This electrical stimulation helps the release of the VEGF, the differentiation of the stem cells to the native cardiomyocytes or the differentiation to the endothelial cells in the angiogenesis processes, improving the hemodynamic processes of the heart, cell replacement and ventricular thickening of the heart. Abbreviations, NHAc = acetylated amino group, NHR = functionalized amino group.

**Table 1 molecules-28-01920-t001:** Description of the function and main findings of the drugs administered during the treatment of myocardial infarction.

Treatment	Drug	Function	Featured Findings	Precautions	Ref.
β-blockers	Metoprolol^®^	Blocks β-1 receptors of cardiomyocytes, reducing blood pressure and muscle contractility	Decreases ischemic damage in people with STEMI. Increases the odds of surviving a second heart attack	Hospitalization in patients with obstructive problems; pulmonary severe or moderate	[51,52,53]
	Carvedilol^®^	Blocks β-1, β-2 and α1 receptors on cells in the body	Helps in the ventricular remodeling processes. Promotes diastolic ventricular filling and decreases ischemic heart disease	Population on hemodialysis could experience cardiovascular complications in the first year of medication	[54,55]
	Nabivolol^®^	β-1 receptor antagonist with vasodilation properties	Well tolerated in elderly patients with heart failure. Decreases the probability of heart failure due to its vasodilator effect	It could cause headaches, diarrhea, nausea, bradycardia and other mild side effects	[56,57]
	Propanolol^®^	Non-specific inhibitor of β-adrenergic receptors	Reduces mortality in post-infarction patients and regulates cardiac arrhythmias	Contraindicated in some patients with grade ≥2 cirrhosis due to a possible increase in chronic kidney damage	[58,59]
	Bisoprolol^®^	Non-specific inhibitor of β receptors. Helps in arterial diminution and vasodilation	It could increase the risk of a heart attack, heart failure and hospitalization vs. β-selective drugs	Hospitalization in type 2 diabetics due to hyperglycemia	[49,60]
Anticoagulants	Rivaroxaban^®^	Decreases thrombotic events by inhibiting serine protease coagulating factor Xa	Helps in the treatment of the left ventricular thrombi after an ischemic episode	Does not significantly reduce the risk of systemic ischemia or embolism in patients with stage 5 renal failure	[61,62]
	Warfarin^®^	Decreases the probability of death, heart attack and systemic embolism by inhibiting coagulation factor II	Helps reduce the likelihood of recurrent heart attacks in people with chronic heart disease	Possible increase in bleeding due to its anticoagulant action	[63]
	Edoxaban^®^	Reduces the risks of ischemia and systemic embolisms through the inhibition of coagulating factor Xa	Prevents ischemia in patients with atrial fibrillation	Adverse liver events in a population with liver problems and atrial fibrillation	[64]
	Betrixaban^®^	Clotting factor inhibitor Xa	Applied in patients with venous thromboembolism	Risk of bleeding	[65]
Antiplatelet	Cangrelor^®^	ATP analog P2Y_12_ receptor antagonist that regulates the platelet segregation	Decreases the probability of a second heart attack. Clinically used to reduce perioperative complications associated with percutaneous coronary interventions	Risk of bleeding	[66,67]
	Voraxapar^®^	Inhibits protease receptor 1, decreasing thrombin synthesis and therefore hindering platelet synthesis	It is administered with P2Y_12_ inhibitor drugs for the biosynthetic decrease in platelets in post-infarction patients	Increases the risk of bleeding	[68,69]
	Clopidrogel^®^	P2Y_12_ receptor antagonist that regulates the platelet segregation	The use of this drug decreases the probability of heart failure and sudden death in long-term patients	Increased risk of bleeding in isolated cases	[70,71]

**Table 2 molecules-28-01920-t002:** Advantages and disadvantages of the chitosan scaffolds incorporating natural and synthetic materials in myocardial infarction tissue treatments.

Composition	Disadvantages	Advantages	Type of Assay	Implantation Site	Animal Model	Cell Type	Ref.
**Chitosan/porcine decellularized extracellular matrix**	Slow cell proliferation within the scaffold of stem cells seeded in the hydrogel	Improves the ejection fractions in rats with acute myocardial infarction and stimulates tissue re-epithelization	In vivo/ In vitro	Left ventricle	Male Wistar rats	-	[106]
**Chitosan/collagen**	Cellular limitation due to low pore connectivity	It would help to up-regulate MicroRNA-1/Myocardin in internally seeded stem cells and reduce the infarcted zone	In vivo/ In vitro	Left ventricle	Male Sprague–Dawley rats	Mesenchymal stem cell	[108]
**Chitosan/PEG/TiO_2_ nanoparticles**	Its rapid biodegradation affects the results of the cell viability studies	This hydrogel presented high biocompatibility, encapsulation and short-term cell binding protein segregation	In vitro	-	-	Neonatal rats cardiomyocytes	[162]
**Chitosan/silk protein**	Activated the hypertrophy process in native cardiomyocytes	Could improve the cellular cardiomyoplasty processes and the hemodynamics of the infarcted heart	In vitro/ in vivo	-	Adult male Sprague–Dawley rats	Bone marrow-derived mesenchymal stem cells	[129]
**Chitosan/graphene oxide**	This material could generate apoptosis in some stem cells	Helps improve cell anchoring and electrical conductivity in the infarcted zone	In vitro	-	-	Cardiomyocytes derived from an engineered human embryonic stem cell line	[163]
**Chitosan/PPi**	Made it difficult to visualize the cells in the cell staining techniques such as hematoxylin and eosin (H&E). The study showed that there was no regenerative capacity of the material.	Improved cell anchoring and electrical conductivity in the infarcted zone, orientation and ordering of the regenerated tissue	In vivo/in vitro	Left ventricle	Female Sprague–Dawley rats	CMs from 1-day-old neonatal Sprague–Dawley rats	[171]
**Chitosan/PPi**	Its application failed to regenerate the infarcted tissue significantly	Used to stimulate the fibrous area of infarction electrically, synchronizing the infarcted tissue and improving the hemodynamics of the heart	In vitro/ In vivo	Left ventricle	Female Sprague-Dawley rats	Fibrotic tissue matrix	[159]
**Chitosan/polyvinyl alcohol/MWCNTs**	Several factors are required for the scaffold to develop functional heart tissue	Scaffold is capable of propagating electrical impulses. Helps induce stem cell differentiation into the cardiomyocytes	In vitro	-	-	Unrestricted somatic stem cells	[172]
**Chitosan/Polyvinyl alcohol/CNTs**	Low levels in the genes that express the differentiation to cardiomyocytes	The scaffold is capable of conducting electricity and with good cell viability values (viability > 80%)	In vitro	-	-	Undifferentiated mesenchymal stem cells from thighbone marrow of young male rats	[84]
**Polyurethane/chitosan/CNTs**	No in vitro/in vivo assay information found	Scaffold can conduct electricity and has good cell viability values in the H9C2 cells	-	-	-	-	[168]
**Chitosan/decellularized tissue of the bovine heart**	The biocompatibility of the scaffold against cardiomyocytes is not known.	Formation of blood vessels inside the scaffold with potential application in the regeneration of cardiac tissue	In vitro	-	-	Human umbilical vein endothelial cell	[173]
**Chitosan/placental insulin-like growth factor 1 C-domain peptide**	Rapid biodegradation in vivo that did not allow for cell maturation	Can improve the hemodynamics of the heart by removing the fibrous tissue in the area of infarction	In vivo/in vitro	Left ventricle	C57BL/6 transgenic mice	Human placenta-derived mesenchymal stem cells/Neonatal rat cardiomyocytes	[109]
**Chitosan/Cardiomyocytes derived from pluripotent human stem cells/decellularized porcine extracellular matrix**	Its application could decrease the capacity of cell maturation	Using this hydrogel improved the concentration of Ca^2+^ and the proteins necessary for the maturation of the stem cells. In addition, it is an isoproterenol-releasing drug.	In vitro	-	-	Human-induced pluripotent stem cell-derived cardiomyocytes	[145]
**Chitosan/graphene quantum dots/collagen**	The hydrogel does not have the necessary capabilities to be injected	The hydrogel could increase the endothelial cells and reduce the infarcted zone in rats, improving the hemodynamics of the treated heart.	In vitro/ In vivo	Left ventricle	Male Fischer rats	Human coronary artery endothelial cells/Human mesenchymal stem cells	[130]
**Chitosan/** **bone marrow-derived mesenchymal stem cells**	The hydrogel does not have the necessary capabilities to be injected into the myocardium	Its superficial application can improve the regeneration of the infarcted tissue with anti-inflammatory effects	In vivo	Left ventricle	Transgenic FVB-Fluc/GFP mice	Bone marrow-derived Mesenchymal stem cell	[148]
**Chitosan/** **Dextran/** **β-glycerophosphate**	It was not possible to measure the regenerative capacity of the hydrogel	In the biomodels, the scaffold demonstrated the stimulation of the differentiation capacity of the mesenchymal stem cells from the umbilical cord to immature cardiac cells	In vitro	-	-	Umbilical cord mesenchymal stem cells	[146]
**Chitosan/Au nanoparticles/Polyglycerol sebacate**	In vitro tests showed a cytotoxicity of 10%	Its composition can allow for the injection of stem cells into the infarcted tissue	In vitro	-	-	H9C2 cells	[174]

Abbreviations: PEG: polyethylene glycol; TiO_2_: titanium dioxide; PPi: polypyrrole; MWCNTs: multi-walled carbon nanotubes; CNTs: carbon nanotubes.

## Data Availability

Not applicable.

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
