# Peer review of "Chitosan-Based Scaffolds for the Treatment of Myocardial Infarction: A Systematic Review"

_molecules, 2023, doi:10.3390/molecules28041920_

Round 1

Reviewer 1 Report

Overall, this is an up-to-date and well-designed review. I applaud that the Authors have taken a methodological approach, applying a well-established PRISMA data collection strategy. Still, it is my strong belief that the study could and should be improved before being published. Please note, that given the short timeframe for reviewing a review study (7-10 days is certainly not enough) my review might not be exhaustive and there still might be more issues present apart from the ones listed below.

1.       The Authors are kindly referred to read some literature regarding the construction of a good quality review article and apply some of the hints given within. For example: https://www.ncbi.nlm.nih.gov/pmc/articles/PMC3715443/. Specifically, I believe that the Authors had failed to perform a critical analysis of the studies they are citing, and instead, they decided to just list the main findings of the articles. While the studies are up-to-date and interesting, some criticism would enhance this review and prove that the Authors do have sufficient background knowledge to be writing a review as such. Just to quote: “Reviewing the literature is not stamp collecting. A good review does not just summarize the literature, but discusses it critically, identifies methodological problems, and points out research gaps”.

2.       Furthermore, while this review focuses on recent articles (from the last 5 years), some more historical background should be painted to better place the current achievements in the broader scope. Were there any other review articles on the matter of chitosan in CVD published? Is there any other material that is promising in this application? What is the current understanding of how a heart can be regenerated? What kinds of cells reside within and how can regeneration be triggered? All of this information should be found in this study but are not at this point. On the other hand, I don’t think that in the review which is focused on the “Treatment of Myocardial Infarction” there is a place for Table 2 listing “Some applications of chitosan in reconstructive tissue engineering”. It is not justified and blurs the subject of the study. I understand that the objective here was to emphasize the chitosan’s advantages but it seems that too much attention and space are devoted to things that are not the subject of the study.

3.       The structure of the article should be altered so that it shows a more clear and more consecutive flow of thoughts. Right now, we have MI, then chitosan, then scaffolds in MI, then bioactivity, then forms of chitosan in MI. Hence, it seems that valid information is scattered throughout the study.

4.       It seems to me that there is a mixture of concepts – hydrogels, patches, and microfilamentous/microporous scaffolds are listed together as scaffold forms. Meanwhile, they concern different things – hydrogel relates to the chemistry of the material, filamentous/porous morphology relates to the form into which the material is processed (i.e., its morphology), while the patch is the mode in which the material is applied. Fibrous scaffolds can be made out of hydrogel materials and applied as cardiac patches. In fact, most of the chitosan materials, regardless of their morphology, are hydrogels. Hence, this is not a proper division of the materials applied in cardiac tissue engineering and needs correcting. The materials should either be grouped by their chemical composition (hydrogels, polymers, etc.), morphology (films, layers, fibrous materials, porous materials, 3D printed materials, etc.), or form of application (tissue engineering scaffold, cardiac patch, injectable materials, etc.). I would also advise introducing division into in vitro and in vivo cardiac tissue engineering materials (D. Sarkar, et al., Chapter II.6.2 - Overview of Tissue Engineering Concepts and Applications, Editor(s): Buddy D. Ratner, Allan S. Hoffman, Frederick J. Schoen, Jack E. Lemons, Biomaterials Science (Third Edition), Academic Press, 2013, ISBN 9780123746269).

5.       The bioactivity of biomaterials is not precisely defined. Citing the Authors, bioactivity “consists of the ability to stimulate cell migration and proliferation, to improve tissue formation in the implantation area. The materials interact with the physiological environment, allowing the release of biomolecules such as proteins, growth factors, and extracellular matrix, counteracting existing apoptosis”. It is unclear, what are those molecules supposed to be released from. Additionally, according to the current and generally accepted definition, bioactivity entails “eliciting a specific biological response at the interface of the material, resulting in the formation of a bond between the tissues and the material.”( Buddy D. Ratner, Allan S. Hoffman, Frederick J. Schoen, Jack E. Lemons, Biomaterials Science (Third Edition), Academic Press, 2013, ISBN 9780123746269). In other words, bioactivity is the ability of the material to develop a strong bond between the tissue, with a minute or negligible fibrous capsule. Furthermore, the article cited does not contain the given definition. Instead, the bioactivity in this study concerns bioactive molecules – which is something entirely different than the bioactivity of scaffolds.

6.       Overall, in some cases, the analysis of the materials is very superficial and does not present some of the outstanding achievements in the field or guidance regarding the direction in which the field should be or is evolving. The subject of biomimicry is barely touched and electrical conductivity is given little attention.

7.       Are graphs in this study self-made by the Authors? If not, citations and permissions are needed.

Some more detailed remarks:

1.       PRISMA abbreviation given in the abstract should be expanded to its full name on its first mention. Later on, an abbreviation can be used.

2.       Proper expansion of the CVD is cardiovascular disease and not cardiovascular complications

3.       Line 51-52. Again, there’s a mistake in the basic definition concerning biomaterials. Tissue engineering, by definition, entails the regeneration (or replacement) of tissues and not their reconstruction (D. Sarkar, et al., Chapter II.6.2 - Overview of Tissue Engineering Concepts and Applications, Editor(s): Buddy D. Ratner, Allan S. Hoffman, Frederick J. Schoen, Jack E. Lemons, Biomaterials Science (Third Edition), Academic Press, 2013, ISBN 9780123746269). Reconstruction can be accomplished by using inert implants that overtake the role of the tissue – and this certainly is not the goal of cardiac tissue engineering.

4.       Line 89. Some more criticism is needed for the analysis of the cited studies. Respected researchers in the field suggest that the cells isolated from the Wharton jelly are not "stem cells" but "mesenchymal stromal cells"  (https://stemcellsjournals.onlinelibrary.wiley.com/doi/full/10.1002/sctm.16-0492) and whether these cells should even be qualified as stem cells is disputable. 

5.       Throughout the study, the Authors raise claims that some materials can “improve the regenerative potential of heart”. Meanwhile, it is now generally accepted that the heart has little to no regeneration potential as it does not contain a stem cell niche (E. Karbassi, A. Fenix, S. Marchiano, N. Muraoka, K. Nakamura, X. Yang, C.E. Murry, Cardiomyocyte maturation: advances in knowledge and implications for regenerative medicine, Nat. Rev. Cardiol. 17 (2020) 341–359. https://doi.org/10.1038/s41569-019-0331-x). Hence, some rephrasing is needed.

6.       Line 169. When the Authors are discussing the electrophysiology (ECG) of the heart and in particular the changes in the ST segment, I think it would be beneficial to show an actual, graphical representation of the data.

7.       The phrase “on the other hand” is excessively overused in the study (for example, lines 223 – 238)

8.       There are some phrase repetitions and article repetitions easily identifiable throughout the study (e.g. line 231, line 250, etc.). Interestingly, in some of the cases where fragments of the text are doubled, different studies are cited (e.g. lines 245 – 254, citation 84 vs 85).

9.       More discussion regarding how can chitosan’s functionalization affect its applicability as cardiac TE materials would be beneficial.

10.   Lines 279 – 300. Again, I don’t see the need for an extensive evaluation of chitosan in skin tissue engineering applications – this is not the subject of the study.

11.   Line 288 – the study Authored by Maharjan is not under number 93 and it does not contain any information about the antibacterial effect of chitosan. Instead, a study by Yang from 2021 contains information about the antibacterial effect of injectable wound healing scaffold. Herein, the bactericidal effect was rather not simply “due to the complex formed”

12.   Line 330 – a fragment of the text is written in Spanish (I guess).

13.   Line 371 – the Authors claim “These problems are addressed by converting native heart fibroblasts to cardiomyocytes”. I would say that such a possibility is rather disputable, so more citations and a thorough discussion are needed. Later on, in situ cellular reprogramming of stem cells in the heart is claimed. Again, as there are no stem cells in the heart, this is impossible (https://www.nature.com/articles/s41569-019-0331-x). Plus, converting stem cells is something entirely different than reprogramming fibroblasts. In theory, one could use fibroblasts as a source of iPSCs and then differentiate these into cardiomyocytes. But this certainly cannot be done in situ.

14.   Line 380 – while this section is titled “chitosan hydrogels”, some of the cited studies concern different materials. Why is that?

15.   Line 393 – a study is cited which is supposed to concern mixing chitosan with PCL, but it does not.

16.   Line 419 – citation 137 is a review article, so I believe that an incorrect study is cited

17.   Line 428 – material cannot “grow cells”

18.   Section 5.3 some of the articles cited in this part concern neither porous nor fibrous/filamentous materials

19.   Lines 451 – 455. The sentences presented herein don’t seem to be correlated. How can electrical stimulation of cells be connected to using chitosan with PLGA or PLLA (non-conductive materials)?

20.   Line 517 – citation number 167 does not concern cardiac tissue engineering.

21.   Line 526 – the first sentence needs rephrasing.

22.   The citation list needs careful evaluation. Some articles are doubled, different styles are used, some studies lack Authors (e.g. 140), and some lack DOIs (e.g. 136). In some cases, incorrect studies are cited – the Authors’ names in the citations do not match the ones that are discussed.

23.   Line 571 -574. This sentence needs rephrasing. 

Author Response

Dear Editor,

We want to submit our revised version of our article entitled "Chitosan-Based Scaffolds for the Treatment of Myocardial Infarction: A Systematic Review." The paper was authored by Bryan Beleño Acosta, Rigoberto C. Advincula, and Carlos David Grande-Tovar, who agreed with all the corrections. The corrections are presented below point by point in red for easy comprehension.

Reviewer 1

Overall, this is an up-to-date and well-designed review. I applaud that the Authors have taken a methodological approach, applying a well-established PRISMA data collection strategy. Still, it is my strong belief that the study could and should be improved before being published. Please note, that given the short timeframe for reviewing a review study (7-10 days is certainly not enough), my review might not be exhaustive, and there still might be more issues present apart from the ones listed below.

R// We are deeply thankful for the reviewer's effort in such a detailed manuscript review. We tried to address point by point all the requirements to improve the manuscript's quality.

  1. The Authors are kindly referred to read some literature regarding the construction of a good quality review article and apply some of the hints given within. For example: https://www.ncbi.nlm.nih.gov/pmc/articles/PMC3715443/. Specifically, I believe that the Authors had failed to perform a critical analysis of the studies they are citing, and instead, they decided to just list the main findings of the articles. While the studies are up-to-date and interesting, some criticism would enhance this review and prove that the Authors do have sufficient background knowledge to be writing a review as such. Just to quote: "Reviewing the literature is not stamp collecting. A good review does not just summarize the literature, but discusses it critically, identifies methodological problems, and points out research gaps".

R// We appreciate the reviewer's suggestions. We have deepened and improved the critical analysis by discussing and comparing the results of each investigation throughout the document.

  1. Furthermore, while this review focuses on recent articles (from the last 5 years), some more historical background should be painted to better place the current achievements in the broader scope. Were there any other review articles on the matter of chitosan in CVD published? Is there any other material that is promising in this application? What is the current understanding of how a heart can be regenerated? What kinds of cells reside within and how can regeneration be triggered? All of this information should be found in this study but are not at this point. On the other hand, I don't think that in the review which is focused on the "Treatment of Myocardial Infarction" there is a place for Table 2 listing "Some applications of chitosan in reconstructive tissue engineering". It is not justified and blurs the subject of the study. I understand that the objective here was to emphasize the chitosan's advantages, but it seems that too much attention and space are devoted to things that are not the subject of the study.

R// We appreciate the reviewer's suggestions. In lines 76-81, we have included the role of chitosan in treating CVDs and the first application of fibrous scaffolds indirectly. This information was entered to contextualize the manuscript. In Lines 248-251, the alginate biomaterial was exposed, and the Algisyl-LVR scaffold used in the treatment of MI was cited. Between lines 261-266, we comment on how the heart can be regenerated, focusing on regenerative therapy.

On the other hand, in the manuscript, we talk about the type of cells used with the chitosan scaffolds. For example, in the section "natural scaffolds with chitosan," lines 371-374, reference 94, Table 2, lines 559, reference 132 and 155, and other sections, more stem cells incorporated with chitosan materials are mentioned. Taking into account the suggestions of evaluator 2 to make the document more dynamic with visual information, we opted to reduce the content in table 2 and leave the most relevant findings discussed between lines 371-388.    

  1. The structure of the article should be altered so that it shows a more clear and more consecutive flow of thoughts. Right now, we have MI, then chitosan, then scaffolds in MI, then bioactivity, then forms of chitosan in MI. Hence, it seems that valid information is scattered throughout the study.

R// We appreciate the reviewer's suggestions. We have modified the sections of the manuscript as follows:

  1. Introduction (Line 37)
  2. Methodology (Line 117)
  3. Myocardial infarction (Line 167)
  4. Scaffolds for cardiac tissue regeneration (Line 260)
    • Scaffold's bioactivity in myocardial tissue engineering (Line 294)
  5. Chitosan (Line 324)
    • Natural polymers/chitosan scaffolds in MI (Line: 390)
    • Synthetic polymers/chitosan scaffolds in MI (Line: 397)
  6. Angiogenesis process (Line 586)
  7. Concluding remarks and perspectives (Line: 646)

  1. It seems to me that there is a mixture of concepts – hydrogels, patches, and microfilamentous/microporous scaffolds are listed together as scaffold forms. Meanwhile, they concern different things – hydrogel relates to the chemistry of the material, filamentous/porous morphology relates to the form into which the material is processed (i.e., its morphology), while the patch is the mode in which the material is applied. Fibrous scaffolds can be made out of hydrogel materials and applied as cardiac patches. In fact, most of the chitosan materials, regardless of their morphology, are hydrogels. Hence, this is not a proper division of the materials applied in cardiac tissue engineering and needs correcting. The materials should either be grouped by their chemical composition (hydrogels, polymers, etc.), morphology (films, layers, fibrous materials, porous materials, 3D printed materials, etc.), or form of application (tissue engineering scaffold, cardiac patch, injectable materials, etc.). I would also advise introducing division into in vitro and in vivo cardiac tissue engineering materials (D. Sarkar et al., Chapter II.6.2 - Overview of Tissue Engineering Concepts and Applications, Editor(s): Buddy D. Ratner, Allan S. Hoffman, Frederick J. Schoen, Jack E. Lemons, Biomaterials Science (Third Edition), Academic Press, 2013, ISBN 9780123746269).

R// We greatly appreciate the reviewer's suggestions. Due to the inconsistency in the past way of grouping biomaterials (hydrogels, porous and filamentous structures, patches), we have modified and regrouped the chitosan scaffolds considering their chemical composition. In the manuscript, it is possible to see the following sections:

5.1          Naturals scaffolds with chitosan in MI   (Línea: 390)

5.2          Synthetic scaffolds with chitosan in MI( Línea: 397)

These sections classify scaffolds by their chemical constitution with proteins, polysaccharides, growth factors, and natural and synthetic polymers. This form of grouping is more general but allows the grouping of the scaffolds without inconsistency. On the other hand, the "Angiogenesis process" is the in vivo/in vitro tests section for materials evaluated in vivo or in vitro. In this section, we studied the properties observed in the tests.

  1. The bioactivity of biomaterials is not precisely defined. Citing the Authors, bioactivity "consists of the ability to stimulate cell migration and proliferation, to improve tissue formation in the implantation area. The materials interact with the physiological environment, allowing the release of biomolecules such as proteins, growth factors, and extracellular matrix, counteracting existing apoptosis". It is unclear, what are those molecules supposed to be released from. Additionally, according to the current and generally accepted definition, bioactivity entails "eliciting a specific biological response at the interface of the material, resulting in the formation of a bond between the tissues and the material." ( Buddy D. Ratner, Allan S. Hoffman, Frederick J. Schoen, Jack E. Lemons, Biomaterials Science (Third Edition), Academic Press, 2013, ISBN 9780123746269). In other words, bioactivity is the ability of the material to develop a strong bond between the tissue, with a minute or negligible fibrous capsule. Furthermore, the article cited does not contain the given definition. Instead, the bioactivity in this study concerns bioactive molecules – which is something entirely different than the bioactivity of scaffolds.

R// We appreciate the reviewer's suggestions. We have modified our definition of bioactivity according to the exposed reference, which can be seen between lines 294-302.

  1. Overall, in some cases, the analysis of the materials is very superficial and does not present some of the outstanding achievements in the field or guidance regarding the direction in which the field should be or is evolving. The subject of biomimicry is barely touched and electrical conductivity is given little attention.

R// We appreciate the reviewer's suggestions. We have deepened the discussion and critique of some studies in the manuscript, i.e., lines 447-488.

  1. Are graphs in this study self-made by the Authors? If not, citations and permissions are needed.

 R// We appreciate the reviewer's suggestions. Each figure in the study is free of copyright since the authors made them with different free access programs.

Some more detailed remarks:

S1.       PRISMA abbreviation given in the abstract should be expanded to its full name on its first mention. Later on, an abbreviation can be used.

R// We appreciate the reviewer's suggestions. Expanded the abbreviation "PRISMA" to Preferred Reporting Items for Systematic Reviews and Meta-Analyses in the abstract, line 22

S2.       Proper expansion of the CVD is cardiovascular disease and not cardiovascular complications

R// We appreciate the reviewer's suggestions. We corrected the abbreviation "CVD" to cardiovascular disease throughout the document.

S3.       Line 51-52. Again, there's a mistake in the basic definition concerning biomaterials. Tissue engineering, by definition, entails the regeneration (or replacement) of tissues and not their reconstruction (D. Sarkar, et al., Chapter II.6.2 - Overview of Tissue Engineering Concepts and Applications, Editor(s): Buddy D. Ratner, Allan S. Hoffman, Frederick J. Schoen, Jack E. Lemons, Biomaterials Science (Third Edition), Academic Press, 2013, ISBN 9780123746269). Reconstruction can be accomplished by using inert implants that overtake the role of the tissue – and this certainly is not the goal of cardiac tissue engineering.

R// We appreciate the suggestion from the reviewer. We modified the information on the basic definition of biomaterials and tissue engineering in the introduction section (lines 52-57).

S4.       Line 89. Some more criticism is needed for the analysis of the cited studies. Respected researchers in the field suggest that the cells isolated from the Wharton jelly are not "stem cells" but "mesenchymal stromal cells"

  (https://stemcellsjournals.onlinelibrary.wiley.com/doi/full/10.1002/sctm.16-0492) and whether these cells should even be qualified as stem cells is disputable.

R// We appreciate the evaluator's suggestions. A more detailed analysis of the highlighted definition was included between lines 97-101.

S5.       Throughout the study, the Authors raise claims that some materials can "improve the regenerative potential of heart". Meanwhile, it is now generally accepted that the heart has little to no regeneration potential as it does not contain a stem cell niche (E. Karbassi, A. Fenix, S. Marchiano, N. Muraoka, K. Nakamura, X. Yang, C.E. Murry, Cardiomyocyte maturation: advances in knowledge and implications for regenerative medicine, Nat. Rev. Cardiol. 17 (2020) 341–359. https://doi.org/10.1038/s41569-019-0331-x). Hence, some rephrasing is needed.

R// We appreciate the evaluator's suggestions. The information in the article has been modified, and with it, the focus of "promoting heart regeneration" has been changed to "helping the renewal of dysfunctional tissue by functional"; i.e., Lines: 274-276., 431-434, 451-457, among others.

S6.       Line 169. When the Authors are discussing the electrophysiology (ECG) of the heart and in particular the changes in the ST segment, I think it would be beneficial to show an actual, graphical representation of the data.

R// We appreciate the reviewer's suggestions. Figure 2a was added to the myocardial infarction section to show the changes in the ST segment

S7.       The phrase "on the other hand" is excessively overused in the study (for example, lines 223 – 238)

R// We appreciate the evaluator's suggestion. The phrase "on the other hand" was changed throughout the text to equivalent phrases.

S8.       There are some phrase repetitions and article repetitions easily identifiable throughout the study (e.g. line 231, line 250, etc.). Interestingly, in some of the cases where fragments of the text are doubled, different studies are cited (e.g. lines 245 – 254, citation 84 vs 85).

R// We appreciate the reviewer's suggestion. Duplicate paragraphs in the document were identified and discarded

S9.       More discussion regarding how can chitosan's functionalization affect its applicability as cardiac TE materials would be beneficial.

R// We appreciate the reviewer's suggestion. Some discussion on the influence of the chitosan's chemical modification on the properties can be observed between lines 440-443 and 574-581.

S10.   Lines 279 – 300. Again, I don't see the need for an extensive evaluation of chitosan in skin tissue engineering applications – this is not the subject of the study.

R// We appreciate the reviewer's suggestion. We have only left the discussion of the bactericidal properties of chitosan made by Yang et al. (2021), and some references that we consider important from table 2 were removed. Lines 356-388 have been reduced.

  1. Line 288 – the study Authored by Maharjan is not under number 93, and it does not contain any information about the antibacterial effect of chitosan. Instead, a study by Yang from 2021 contains information about the antibacterial effect of injectable wound healing scaffold. Herein, the bactericidal effect was rather not simply "due to the complex formed"

R// We appreciate the reviewer's suggestion. Reference number 122 corresponded to Yang (2021), not to Maharian. We corrected the citation. Lines 361-370.

  1. Line 330 – a fragment of the text is written in Spanish (I guess).

R// We appreciate the reviewer's suggestion. The snippet was correctly written in English (lines 287-291).

  1. Line 371 – the Authors claim, "These problems are addressed by converting native heart fibroblasts to cardiomyocytes". I would say that such a possibility is rather disputable, so more citations and a thorough discussion are needed. Later on, in situ cellular reprogramming of stem cells in the heart is claimed. Again, as there are no stem cells in the heart, this is impossible (https://www.nature.com/articles/s41569-019-0331-x). Plus, converting stem cells is something entirely different than reprogramming fibroblasts. In theory, one could use fibroblasts as a source of iPSCs and then differentiate these into cardiomyocytes. But this certainly cannot be done in situ.

R// We appreciate the reviewer's suggestion. We modify the statement highlighting that the reprogramming of fibroblasts to cardiomyocytes is used to obtain induced cardiomyocytes from induced-pluripotent Stem cells. The new information is as follows: "These problems are addressed by converting native heart fibroblasts to induced pluripotent stem cells and finally induced-cardiomyocytes with transcription factors, growth factors, specific biological stimuli, or 3D cardiac reconstruction with biocompatible scaffolds through in vitro cellular reprogramming." (lines: 406-410).

  1. Line 380 – while this section is titled "chitosan hydrogels," some of the cited studies concern different materials. Why is that?

R// We appreciate the reviewer's suggestion. We have modified the "hydrogels" section and have included sections 5.1 and 5.2 (lines: 390 and 497, respectively). With this new categorization, we generalize the grouping of information, and we can present hydrogels with other materials.

  1. Line 393 – a study is cited which is supposed to concern mixing chitosan with PCL, but it does not.

R// We appreciate the reviewer's suggestion. Studies related to PCL were correctly referenced (reference 138; Line 497).

  1. Line 419 – citation 137 is a review article, so I believe that an incorrect study is cited

R// We appreciate the reviewer's suggestion. The reference was modified, and the reference of the corresponding study was placed (Reference 141, line: 507).

  1. Line 428 – material cannot "grow cells"

R// We appreciate the reviewer's suggestion. We have changed the term cell growth to "or differentiation of exogenous seeded-cell in regenerative therapies" Since the materials cannot help or promote the division of cardiomyocytes, therefore, we improve the discussion by criticizing the processes of hemodynamic improvements with little follow-up micro-scale effects of implanted exogenous cells (Lines 533-535).

  1. Section 5.3 some of the articles cited in this part concern neither porous nor fibrous/filamentous materials

R// We appreciate the reviewer's suggestion. We have modified the "hydrogels" section and included sections 5.1 and 5.2 (lines: 390 and 497, respectively). With this new categorization,  we believe that the information makes more sense.

  1. Lines 451 – 455. The sentences presented herein don't seem to be correlated. How can electrical stimulation of cells be connected to using chitosan with PLGA or PLLA (non-conductive materials)?

R// We appreciate the reviewer's suggestion. Reference 144, related to PLGA and PLLA, was removed due to an inconsistency (lines 544-547).

  1. Line 517 – citation number 167 does not concern cardiac tissue engineering.

R// We appreciate the evaluator's suggestion. The reference was modified, and the original reference "Jing, X.; My, H.-Y.; Napiwocki, B.N.; Peng, X.-F.; Turng, L.-S. Mussel-Inspired Electroactive Chitosan/Graphene Oxide Composite Hydrogel with Rapid Self-Healing and Recovery Behavior for Tissue Engineering. Carbon N.Y. 2017, 125, 557–570, doi:10.1016/j.carbon.2017.09.071.” (line: 598-602).

  1. Line 526 – the first sentence needs rephrasing.

R// We corrected the sentence (lines 602-607).

  1. The citation list needs careful evaluation. Some articles are doubled, different styles are used, some studies lack Authors (e.g. 140), and some lack DOIs (e.g. 136). In some cases, incorrect studies are cited – the Authors' names in the citations do not match the ones that are discussed.

R// We appreciate the reviewer's suggestion. We have revised the DOI and reference styles, which have been fully corrected.

  1. Line 571 -574. This sentence needs rephrasing.

R// We corrected the paragraph (lines 657-660).

Reviewer 2 Report

Acosta and co-authors presented a Review entitled "Chitosan-Based Scaffolds for the Treatment of Myocardial Infarction: A Systematic Review" for the publication on Molecules. The presented systematic review is well organized, and quite complete, considering the small time range chosen by the authors. I suggest to accept the manuscript after some minor revision.

In detail, I suggest to include more figures in the main text to lighten the reading, perhaps taking a cue from the described papers . Moreover, it would be interesting to add a short section before or within "conclusions" section in which the authors give their expert opinion on the topic.

Author Response

Reviewer 2

Acosta and co-authors presented a Review entitled "Chitosan-Based Scaffolds for the Treatment of Myocardial Infarction: A Systematic Review" for the publication on Molecules. The presented systematic review is well organized and quite complete, considering the small-time range chosen by the authors. I suggest accepting the manuscript after some minor revisions.

R// We sincerely appreciate the positive comments of the reviewer. We have tried to improve the quality of the manuscript for a better understanding and flow of information.

In detail, I suggest including more figures in the main text to lighten the reading, perhaps taking a cue from the described papers.

R// We appreciate the reviewer's suggestion. A new figure has been added to the myocardial infarction section

Moreover, it would be interesting to add a short section before or within the "conclusions" section in which the authors give their expert opinion on the topic.

R// We appreciate the evaluator's suggestion. This section has added a more critical discussion of chitosan scaffolds today (lines 663-679).

Round 2

Reviewer 1 Report

While I am pleased with the way the Authors had approached this review, significantly improving its quality, there are still some major points that need addressing before the study could be regarded as publishable.

1.       I performed a random analysis of some of the cited studies. Those that were investigated appear to be presenting different conclusions and/or applications that what the Authors are suggesting. I am not going to analyze all the cited studies to verify if this is a reoccurring pattern, but this observation is alarming to me. Hence, my advice is for the Authors to double check if they are not reporting some misleading and/or far-fetched information.

2.       Abbreviations, such as PVA, PGA, TTEO (and many others raised in the cited studies) should be expanded on their first mention in the text.

3.       Still, little to no criticism towards the reported studies can be found in this review.

4.       The Authors claim to have reorganized and rewritten the sections so that materials are grouped "considering their chemical composition” (sections 5.1. and 5.2.). It appears to me that this is mostly a change in name, as section 5.2 still concern mostly fibrous scaffolds.

5.       Lines 54, 58 – term “reconstructive tissue engineering” should be avoided as it is imprecise and confusing. By definition, tissue engineering is aimed at regeneration and not reconstruction. As I said before, reconstruction can be accomplished by using inert implants that overtake the role of the tissue – and this certainly is not the goal of cardiac tissue engineering which is supposed to induce tissue healing. Instead, the goal of tissue engineering is to fully restore the native tissue to the state it was in before the disease has happened. 

6.       Line 65. This sentence: “The ease with can be chemically modified” needs rephrasing.

7.       Line 70 – 71. The following phrase needs to be rewritten: “applied for the regeneration of dermal tissue [23], soft tissue [21], and even to heal skin wounds [24]”. Skin is an example of soft tissue and is a dermal tissue, so it makes no sense to introduce division as such.

8.       Line 80 – 91. The sentences from this paragraph need rephrasing as right now they state: “The lack of an aggressive immune response is possible due to the ability to modify the chemical composition (…) since the main requirement is to have mechanical properties similar to those of the myocardium without causing high tension in the walls of the heart or aggressive immune responses. However, chitosan-based scaffolds have a limitation since they do not have mechanical properties similar to the biological tissue of the heart, limiting their application in the field of cardiac engineering.” So, first the Authors state that chitosan does not cause an aggressive immune response because it has good mechanical properties and des not cause immune response (?), just to say in the following sentence that it does not have sufficient mechanical properties.

9.       Lines 96-98. As I already stated in previous review, more criticism should be applied to the cited studies. Ability of Mesenchymal stromal cells derived from Wharton jelly to differentiate into mature cardiomyocytes is extremely doubtful.

10.   Lines 104-106 – the sentence needs correcting.

11.   The exclusion criteria seem a little vague to me. In line 135, it is written: “This review considered including systematic review documents”, while later, in line 148 it is stated that systematic reviews were excluded. So, where these included or not? I assume that the Authors meant that systemic reviews which did not concern chitosan were excluded but this is not clear from the given text.

12.   Line 149, the following phrase needs rewriting: “clinical trials without participating in chitosan scaffolds”

13.   Line 209 what are “collagen-like proteins”?

14.   Line 247, the following sentence needs rephrasing: “Alginate is widely used in bioengineering due to its advances results in clinical trials for different diseases”

15.   Line 251, 252, over usage of “which allow”

16.   Line 259 term “regenerative factor” seems vague.

17.   Line 265, the sentence is missing an object “three-dimensional scaffolds capable of simulating the biological environment and supplying nutrients to stem cells introduced inside have been used”. It is unclear what does the “introduced inside” fragment relate to.

18.   Line 270 – 272, the sentence needs rephrasing as it starts with “among the biocompatible compounds” and ends with “among others”.

19.   Line 272 – 274 needs rephrasing as it starts with “These polymers help by(...) drug and cell release” and ends with “in some cases, are drug-releasing that enhances regenerative processes”.

20.   Line 275 – 277 – the same mixture of concept which was raised in my previous review is retained. Just to quote: “– hydrogels, patches, and microfilamentous/microporous scaffolds are listed together as scaffold forms. Meanwhile, they concern different things – hydrogel relates to the chemistry of the material, filamentous/porous morphology relates to the form into which the material is processed (i.e., its morphology), while the patch is the mode in which the material is applied. Fibrous scaffolds can be made out of hydrogel materials and applied as cardiac patches. In fact, most of the chitosan materials, regardless of their morphology, are hydrogels.”

21.   Lines 279 – 281 – the sentence needs rephrasing

22.   Line 283 verb is missing from the “electrical stimulation”.

23.   Line 292 – 295 I would prefer if the Authors used their own words instead of copy-pasting what I have written.

24.   Lines 305 – 307. The sentence needs rephrasing. ”Chitosan hydrogels with synthetic polymers and stem cells are used for their ability to encapsulate cells”. Stem cells are used for their ability to encapsulate cells? Later, there is no such thing as “stem cell hydrogels”.

25.   Line 309 – 310. The study cited as [110] does not contain any information regarding the vascular endothelial growth factor.

26.   Lines 313 – 315. More insight is needed for the study cited as [112]. The Authors claim that “porous or filamentous chitosan scaffolds, together with gelatin and fibronectin, are essential for differentiating multipotent and pluripotent cells”. In this article, chitosan is not used as a scaffold and its role is entirely different than what the Authors are claiming herein.

27.   Study cited as [113] concerns chitosan/gelatin and chitosan/cardiac ECM covered on the surface of 3D-molded PCL. Referring to these as “chitosan and polycaprolactone cardiac patches (…) reported to improve the synchronization of necrotic and functional cardiomyocytes” is an inaccurate oversimplification. It is a valid question to ask, whether the positive outcomes of the cited study were due to presence of chitosan or rather – peptides/proteins. In fact, in some of the previous studies by Pok et al., a following sentence can be found: “cardiomyocytes do not attach and survive on pure chitosan scaffolds” (https://doi.org/10.1016/j.actbio.2012.10.032).

28.   Lines 322 – 325 – commas are missing from the sentence.

29.   Lines 326 – 329. The sentence needs rephrasing, as it is hard to follow what the Authors had in mind: “Therefore, it is essential in any research that uses chitosan to determine the degree of deacetylation (…) since they (…)  influence its biological behavior [115], widely used to obtain chitosan-derived scaffolds applicable in cell regeneration.”

30.   Line 336. The following needs rephrasing “Substitutions of hydroxyl groups by etherification reactions”. It should be clearly stated what are the hydroxyl groups substituted with.

31.   Reaction from Figure 3 which is described as “oxidation” seems to be rather a crosslinking. Could the Authors elaborate? Furthermore, I would advise that the Figure’s caption identifies in what articles can the presented reactions be found.

32.   Lines 357 – 359. As already stated in my previous review, antibacterial properties of the material presented in the study by Yang is certainly not “due to the complex formed by carboxylated polyvinyl alcohol/ε-poly(L-lysine) and chitosan using oxidized dextran as a crosslinking agent”. Furthermore, the Authors state “silver nanoparticles that improved the bactericidal and regenerative factor of the scaffold 15 days after implantation in skin tissue”. In this study, bactericidal properties of the scaffolds were not tested in vivo.

33.   Line 372. In the cited study, biomodels are used as an alternative description of rats. The way this paragraph is written makes it hard to grasp what does this refer to herein. Furthermore, phrase “CaCl2 and NaCl capsules form in the slides during their exposure to a simulated biological fluid” is imprecise, as “capsules” may suggest formation of particles, wherein, the study suggest that a thin layer of salts was formed surrounding the material. What is more, a conclusion that formation of a layer of salts indicates bioactivity is far-fetched – more criticism on the Authors’ side should be employed. Later on, the Authors suggest that “These investigations reinforce that chitosan will obtain acceptable immunological responses regardless of the organ where it is implanted”. Again, this is a far-fetched conclusion – the fact that the material performs acceptable in the spinal cord and upon subcutaneous implantation does not indicate that it will perform satisfactory in other applications. What is more, the cited studies do not concern pure chitosan, but its blends, making such conclusions even more underminable.

34.   Overall, in the whole section 5.1. some unnecessary repetitions can be found, and the important information seems to be mixed and matched throughout the section. It is really hard to follow the flow of thoughts here. For example, information regarding biocompatibility or biomimicry to glycosaminoglycans. Porosity is also mentioned in three different places. First, in line 426, then in line 440, and finally, in two last paragraphs - these start talking about porosity, but the importance of the reported results remains unclear. There is also an oversimplification in line 485, which claims that chitosan and other natural polymers resemble native glycosaminoglycans – this is clearly not true for some natural polymers, such as peptides. Furthermore, how exactly do the Authors envision that the non-conductive materials as such can “optimize the electrical communication of infracted or necrotic muscle”?

35.   Lines 387 – 390. The connection between the first and the second sentence is elusive.

36.   Lines 391 – 394 – citation is missing. 

37.   Lines 401 – 404 converting fibroblasts into iPSCs and reprogramming them into cardiomyocytes does not address the issue of small cellular survival at the implanted site. In fact, these cardiomyocytes might also suffer from apoptosis and poor anchorage, while also presenting immature phenotype. These section needs correcting, clarifying, and rewriting.

38.   Line 410. The cited study does not analyze the chitosan performance in the body, only in vitro results are given. Furthermore, some more criticism should be applied regarding the way the Authors of the cited study have evaluated the stem cells differentiation (and whether these are indeed stem cells).

39.   Line 414 – the sentence needs rewriting.

40.   Lines 417 – 419. Some clarification is needed, first it is mentioned that chitosan/protein blends have rapid biodegradability, and then, that incorporation of proteins improves the biodegradability. It should also be stated what do the Authors mean as ‘improved’ herein.

41.   Line 427 what do the Authors have in mind while saying “at least the scaffold biodegradability increased”?

42.   Line 433 – there is no such thing as “cell diameter pore”.

43.   Lines 439 – 442. The connection between the first and the second sentence is elusive.

44.   Line 440 – the cited study does not really evaluate the cells’ viability but their morphology upon injection. There is no claim that the viability of UCMSCs is lower in the tested scaffold than in the control, and the Authors of the cited study make no connection with materials’ porosity.

45.   Line 442 – 447. It seems elusive to me how exactly is this fragment connected with the discussion about chitosan/dextran composite.

46.   Lines 465 – 470 need rewriting. The Authors cite the studies indicating the materials’ excellent properties only to then claim that “chitosan scaffolds and natural biomaterials have shown low therapeutic properties in vivo”. Furthermore, the Authors literally claim that “chitosan materials are antioxidants and that is why they have been reported to have antioxidant properties”. Later on, it is written that antioxidant properties are not native to these materials but are attributed to some modifications. It is really hard to follow what was the intention of this paragraph. Some commas are also missing in these sentences.

47.   Overall, section 5.2 is written in a similar style to section 5.1, wherein important information are scattered throughout the text, making it hard to understand the take-away message, or what the Authors actually had in mind. For example, introduction of electrically conductive additives is first mentioned in line 513, then, again in line 549, and finally, in line 575. The study would be much easier to read if particular approaches were grouped together.

48.   Lines 504 – 506 citation is missing at the end of paragraph.

49.   In general, when Authors are raising that the electrical conductivity of scaffolds improves their properties, it should be discussed why exactly is it so.

50.   Line 508 – 509. The sentence needs rewriting.

51.   Line 514. A generally accepted abbreviation of carbon nanotubes reads CNTs (and SWCNTs, MWCNTs).

52.   Lines 519 – 523. The fact that chitosan scaffold can promote the electrical stimulation of cells signaling certainly has nothing to do with the fact that “its production is linked to the use of synthetic polymers such as γ-poly(glutamic) acid, poly(L-lactic) acid, and polyethylene glycol, among others”. This sentence makes no sense and needs to be rewritten.

53.   Line 523, again, there is a mixture of concepts – hydrogels and fibers.

54.   Line 523.  The Authors claim that “Fibrous scaffolds, contrary to hydrogels, fulfill the function of serving as structural and stimulating support for stem cells in regenerative therapy”. Meanwhile, in the previous chapter, it was suggested that even injectable materials can provide sufficient support for the cells to grow on.

55.   Line 529 – 534. This section needs rewriting. E.g. sentences “adding chitosan to 2 wt. % to a fibrous scaffold” or “The microfilaments were formed from the chitosan polymer functionalized through the amino groups of chitosan with the carboxyl groups of the carbon nanotube” make no sense.

56.   Line 543. A study by Assuel [152] is cited even though it does not concern the MI tissue regeneration (it is about development of Small-Diameter Vascular Grafts).

57.   In Table 2, more details regarding “type of assay” should be given – at the very least, cell type and animal model used (both the certain specie and the implantation site).

58.   Line 567 - 574. The Authors claim “Unlike the other structures, the porous chitosan scaffolds can absorb extracellular matrix components better” - better than what? Isn’t a more thorough discussion regarding type of porous structure, morphology of the scaffolds in place here? Further, how exactly can chitosan-based scaffolds “mimic electrical conductivity, distribution of nutrients on the tissue, and constitution of proteins such as extracellular matrix collagen in regenerative processes”? This calls for more elaboration, as currently is imprecise and incorrect.

59.   Line 575 “anomaterials”

60.   Section 6 reads “Angiogenesis processes”, while it covers mostly the electrical stimulation of cells with scaffolds. This should be explained and more thoroughly elaborated on.

61.   Lines 590 – 594. What is the actual advantage of chitosan over other materials in this application? Most of polymers and biopolymers can be synthesized to reach certain morphologies and “organization”. As a matter of fact, chitosan is not particularly malleable, and its processing can be challenging. 

62.   Lines 595 – 600. This fragment seems to belong to section 5.1.

63.   Line 601, the sentence needs rephrasing. It’s the knowledge about this phenomenon recent and not itself.

64.   Some parts of section between 623 – 634 were already mentioned in previous chapters (536 – 545).

65.   Line 655 – 659. The following need rewriting to avoid unnecessary repetition and provide a clearer message: “some materials did not show regenerative activity when used in vivo and without being loaded with stem cells. Additionally, chitosan scaffolds are advancing in efficacy but must be used with stem cells”.

66.   Lines 658, 660 – over usage of “additionally” and “in addition”.

67.   Lines 675 – 677. The Authors state that “In the future, the improvement of this technology for incorporating stem cells and regenerative stimulation of the myocardium is expected, offering a friendlier and less invasive alternative therapy for patients.” Invasiveness and/or unfriendliness of these therapies are not mentioned in the article, so more elaboration is needed herein.

68.   The Author contribution section was altered – formal analysis and project administration were replaced by validation, resources, visualization and funding acquisition (two Authors are listed even though only one funding source for one Author is acknowledged). B.B.A. was added to the methodology contributions. Such changes should be justified.

69.   There’s still an inconsistency in the way citations are presented – is some cases the title of the study is written in italics, in other: the journal name. This should be unified.

Author Response

We appreciate all the reviewer's comments and suggestions and tried to address them point by point. The corrections can be followed in the letter of the attached answer and the manuscript highlighting with blue all the changes performed. 
